# MULTIMODALITY AS SUPERVISION: SELF-SUPERVISED SPECIALIZATION TO THE TEST ENVIRONMENT VIA MULTIMODALITY

**Kunal Pratap Singh**[*]    **Ali Garjani**[*]    **Rishubh Singh**    **Muhammad Uzair Khattak**    **Efe Tarhan**

**Jason Toskov**    **Andrei Atanov**    **Oğuzhan Fatih Kar**[†]    **Amir Zamir**

Swiss Federal Institute of Technology Lausanne (EPFL)

https://tst-vision.epfl.ch

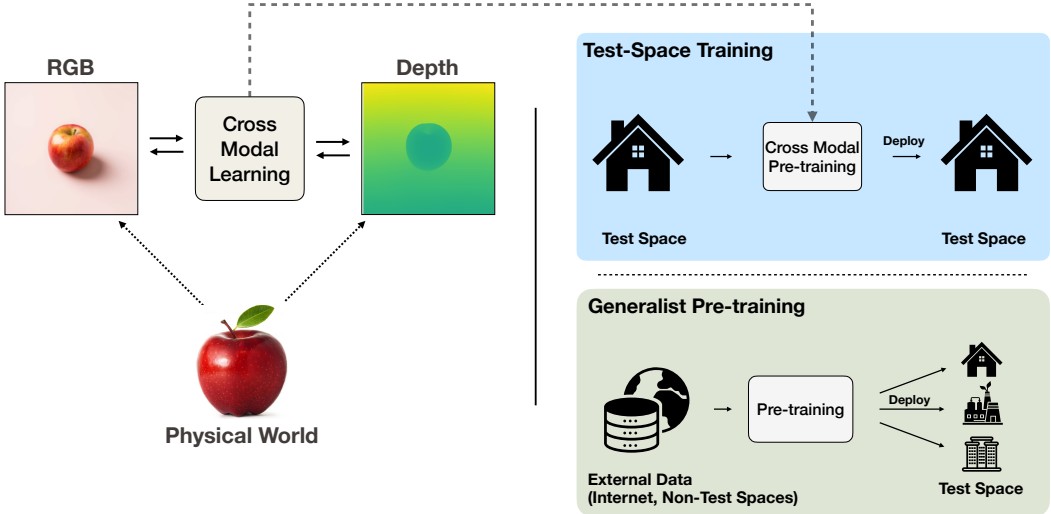

Figure 1: *Left:* **Multimodality as Supervision.** The sensed data in a deployment environment is often multimodal, which, besides RGB images, can contain various modalities, such as depth, motion sensing, surface normals, tactile, etc. This enables *Cross-Modal learning*, i.e., predicting the response of one sensor from another, as a method for *self-supervised* pre-training. We use this concept to frame learning a self-supervised representation for the test space via multimodality.
*Right:* **Test-Space Training.** To study cross-modal learning in a controlled manner, we construct a *specialization* setup, where we perform *self-supervised pre-training* on unlabeled data and evaluation in the same space. This is an alternative to generalist pre-training, which uses large, diverse external data, such as images from the Internet or other external spaces. Our proposed specialization framework *Test-Space Training* (TST) with cross-modal learning outperforms generalist pre-training baselines, including those trained on large-scale Internet-based datasets (Bachmann et al., 2024; Oquab et al., 2023; Radford et al., 2021) or many other external spaces.

## ABSTRACT

**Cross-modal learning**, i.e., learning to predict one modality from another, is a fundamental mechanism for self-supervision via leveraging multimodality. Many practical applications, e.g., deploying a household robot, involve devices that are equipped with a rich set of sensors that enable multimodal sensing in their test environment. This presents an opportunity to apply cross-modal learning to the multimodal data sensed by these devices to learn representations. Findings in developmental psychology also suggest that biological agents leverage it to build an effective representation of their surroundings.

---

[†] Work done while at EPFL. Now at Apple.

To study this, we propose a sandbox, where we restrict a user device to just a given test environment. It results in a **specialization** setup where we attempt to develop a performant model for this specific test environment. Under this setup, we develop Test-Space Training (`TST`), which performs multimodal data collection in the test environment and performs self-supervised pre-training on it. We evaluate these models on various downstream tasks in the same environment. We find various interesting insights, such as collecting *rich multimodal data* only from the test environment and leveraging *cross-modal learning*, we can achieve competitive results with generalist models (Oquab et al., 2023; Radford et al., 2021), pre-trained on large-scale internet-based datasets. This enables an alternative scenario where the need for external Internet-scale datasets for pre-training models is reduced. We also present a set of analyses and ablations that raise intriguing points on *substituting data with (multi)modality*, and how varying pre-training data enables a tradeoff between a model's abilities to specialise to a test environment, and generalize to held-out spaces.

# 1 INTRODUCTION.

**Multimodality is a fundamental self-supervision mechanism**. Consider this, you are a newly born agent with multiple sensory modalities, e.g., sight and sound, but no prior representation for what they represent. The system is in a clean-slate state. The two modalities provide you with two time-locked streams of information sensed from one underlying physical world. By merely learning to **predict one modality from the other**, you can start building a representation for these modalities. This concept is known as "**cross-modal learning**" (Fig 1, left) and requires no external supervision beyond the sensory modalities of the agent.

Applying this concept across a **set of diverse modalities** (sight, sound, haptic, motion, self-action, imitation, smell, etc.), combined with **well-designed pre-training objectives**, can lead to a rich representation without any external supervision, achieved solely through multimodality. Evidence in developmental psychology suggests it as one of the key drivers behind efficient yet effective biological intelligence (Smith & Gasser, 2005). Many everyday user devices that deploy AI, e.g., an iPhone, are already equipped with a rich array of sensors, making this concept readily applicable in the real-world.

In this work, we are interested in studying how far this concept can go in learning vision representations with modern machine learning tools. There are various ways for instantiating this study. Prior work (Bachmann et al., 2022; 2024) has studied this by performing cross-modal learning on large-scale internet-based data (Changpinyo et al., 2021), and shown impressive any-to-any multimodal modelling and downstream generalization via transfer learning on various benchmarks.

We take an alternative approach and adopt the following: we spawn an agent with multiple sensors in one building, or as we call it, *the test space*. The agent moves around in this space and collects data from its sensors. [1] We then perform self-supervised pre-training with cross-modal learning using the process described above on this multimodal data. After pre-training, we test whether the learned representation can be efficiently transferred to novel tasks in this space (e.g., semantic segmentation from images in the test space). We refer to this framework as *Test-Space Training* (`TST`), and describe it in more detail in Sec. 3. In other words, if we assume the entire world of the agent is this one building, we test whether cross-modal learning can yield a representation with emergent properties that can be conveniently transferred to tasks not directly answered by the sensed modalities. This yields a "**specialization**" setup (Fig. 1, right) in which a performant model is expected to be developed for a specific test space. Note that, as the setup is self-supervised, no task-specific label supervision is leaked, thereby making testing and pre-training in the same space valid.

This instantiation setup has three key advantages. First, it provides a *controlled sandbox* to study the question of interest. It brings the world under our control, allowing us to control for data size and diversity, analyze the role of multimodality (Sec. 4.4 and identify the role of pre-training

---

[1] We experiment with two setups. One based solely on physical sensory modalities (Sec. 4.2) and one that expands to use features of pre-trained neural networks as additional modalities (Sec. 4.3). We discuss them in more detail later.

data source on specialization (Sec. 4.5). Second, as opposed to prior experimental setups (Mizrahi et al., 2023; Bachmann et al., 2024; 2022) with internet-scale data, this setup closely resembles the developmental settings of babies and biological agents that inspired the formulation of cross-modal learning. They manage to develop a rich and efficient representation of their surroundings, from the physical context they exist in (Smith et al., 2011; Pereira et al., 2014), in contrast to leveraging diverse data from the entire world to build a generalizable representation that works even outside their surroundings. Lastly, it has **real-world deployment value**. Our setup represents the deployment scenario of many physical agents. Various user devices, such as a household robot, AR/VR goggles, and domestic digital assistants, often do not leave the house where they are deployed. Hence, as long as they perform effectively in that specific house, it does not matter whether it generalizes to other houses in the world. In other words, many cases of deployed AI have **a limited operating context around them**, which is often at odds with the approach of building **"generalist"** agents that are **expected to solve all problems everywhere**. This also relates to a broader point in biology on excessive generalism (De Waal, 2016) which emphasizes that both humans and animals show nuanced context-dependent behaviors.

A central design choice in this study is which modalities to include in the dictionary, as it will directly affect the richness of the learned representation. We first experiment with a bare setup (Sec. 4.2) that includes only modalities easily acquired from hardware sensors, such as RGB images, and depth. We find that even this bare set is effective as it covers almost half of the spectrum between the two bounds of no-pretraining (scratch) and a fully-supervised model (Fig. 11) in the test space. However, this model falls behind state-of-the-art internet-based generalists such as DINOv2 (Oquab et al., 2023). This suggests that the bare set of modalities, given the current model architecture and training objectives, is sub-optimal for replacing generalist models in deployment. We therefore extend the modality dictionary further (Sec. 4.3) to include pseudo-modalities that are latent representations of powerful pre-trained neural networks, such as CLIP (Radford et al., 2021), Imagebind (Girdhar et al., 2023). Leveraging these modalities essentially corresponds to **distilling the pre-trained models and specializing them to the test space**. The multimodal representation trained with this extended set of modalities yields state-of-the-art results in the test space (Tab. 1), and, interestingly, outperforms all pseudolabel networks from which the latent representations (App. N) were distilled.

This raises an interesting question: *whether for scenarios where the deployment space is known, serving off-the-shelf generalist models trained on internet-scale datasets is the most viable solution.* It suggests a possibility that distilling powerful pseudolabels on just deployment space data can be sufficient for performant models[2]. Additionally, beyond performance, specialization may also improve deployment efficiency, since the one-time cost of test-space pre-training can be amortized when smaller specialist models replace larger generalists at inference (App. B).

Through various analyses and ablations, we contribute the following key findings:

• **Multimodality as (Self-)Supervision is effective**, specifically under a test-space specialization setting. Our experiments show that this approach is promising, and when augmented with modalities based on pre-trained models, it leads to state-of-the-art results in the test space. (Tab. 1)

• **Scaling modalities in the test space can substitute external data.** We find that scaling the number of modalities in the test space is more effective (Fig. 4) than scaling unimodal data from an external source.

• **Specialization-Generalization Tradeoff.** We explore how, with the same number of samples, the source of pre-training data can enable a model to exhibit varying abilities, from being specialized to a test space, to generalizing over a set of held-out spaces (Fig. 9).

## 2 RELATED WORK

**Self-supervised learning (SSL)** has been effective in learning visual (et al., 2020; He et al., 2021; Bao et al., 2022; Oquab et al., 2023) and natural language (Devlin et al., 2019; Brown et al., 2020; OpenAI, 2023) representations. In vision, one line of work uses masked image modeling as a scalable approach to pre-train self-supervised models. It masks an input image,

---

[2]Note that this setup does not correspond to a case in which no information or data beyond the test space was seen (see Sec. 4.3 for details)

and attempts to reconstruct it in the form of pixels (He et al., 2021; Chen et al., 2020a; Doso-vitskiy et al., 2021; El-Nouby et al., 2021a), tokens (Bao et al., 2022) or features (Zhou et al., 2022; Baevski et al., 2022). On the other hand, approaches like SimCLR (Chen et al., 2020b) and DINOv2 (Oquab et al., 2023) use contrastive learning (Caron et al., 2021; Chen et al., 2020b; He et al., 2020; Chen & He, 2020) and knowledge distillation (Buciluǎ et al., 2006; Hinton et al., 2015) respectively, to pre-train representations. Both classes of SSL pre-training approaches are typically trained on large-scale Internet-based datasets (Changpinyo et al., 2021; Deng et al., 2009; Schuhmann et al., 2022; Gadre et al., 2024) and exhibit remarkable downstream generalization abilities. In contrast to the self-supervised learning works discussed above, which aim to pre-train on a large-scale dataset to generalize to a variety of downstream scenarios, we are interested in studying multimodality as a source of self-supervision in a specialization setup. We do so by restricting our pre-training and downstream evaluation space to a single test space, and performing cross-modal learning on the multimodal data collected in that space.

**Multimodal learning** aims to build models that can relate information from different sources of underlying reality (Baltruvsaitis et al., 2017). This can involve training separate encoders or a unified model on various sources of modalities, like image, video, 3D, text, audio, etc. (Arandjelovic & Zisserman, 2017; Lu et al., 2019; Jaegle et al., 2022; Radford et al., 2021; Girdhar et al., 2022; Lu et al., 2023b;a; Girdhar et al., 2023). MultiMAE (Bachmann et al., 2022) uses multimodal masked modeling to learn cross-modal predictive coding across multiple modalities. 4M (Mizrahi et al., 2023; Bachmann et al., 2024) extends this idea further to train a multimodal foundation model across tens of modalities. These approaches build on large-scale Internet datasets with image-text pairs (Changpinyo et al., 2021; Byeon et al., 2022; Schuhmann et al., 2022). In this work, we consider an alternative specialization setup wherein we leverage cross-modal pre-training only on deployment space data to pre-train a vision model, just for that space. We show the value of this specialization over internet-based generalists in Sec. 4, by comparing performance over various downstream tasks in the deployment space.

**Pre-training data** has been a critical component in the recent advances in self-supervised (Oquab et al., 2023; Caron et al., 2021) and multimodal foundation models (Mizrahi et al., 2023; OpenAI, 2023; Ramesh et al., 2021). Prior work (Gadre et al., 2024; Li et al., 2024b; Fang et al., 2023) has studied the role of pre-training data curation criteria in large-scale image-text models (Radford et al., 2021). Similarly, (El-Nouby et al., 2021b) investigates the role of external pre-training data for downstream performance on various target transfer tasks (Lin et al., 2014; et al., 2017). They show that, pre-training via masked modelling directly on just the target task images performs on par with pre-training on a large-scale external dataset such as Imagenet (Russakovsky et al., 2014). Our work also similarly sheds light on the importance of pre-training source in building models specialised for a particular test space (Sec. 4.5), as opposed to prior work, which focused on generalization abilities.

**Test-time adaptation** adapts a model to distribution shifts at test-time (see (Xiao & Snoek, 2024) for a recent survey). One prominent approach in the community is test-time training (TTT) (Sun et al., 2020; Wang et al., 2021; Liu et al., 2021b; Gandelsman et al., 2022b; Boudiaf et al., 2022; Gao et al., 2023), which optimizes a self-supervised objective at test-time to finetune the model. Contrastingly, as opposed to optimizing for a specific test instance, we focus on learning a vision model for a given test space *during pre-training* and not on model adaptation *at test-time*. Concretely, we specialize in a given test space, as opposed to a specific test instance. Note that TTT can be complementary to `TST` and improve performance (see App. T). We present additional related works in App. A on domain adaptation, embodied active learning, and semi-supervised learning.

## 3 METHOD

The objective of our work is to study the potential of multimodality as a signal of self-supervised learning. We propose to do so in a controllable sandbox setup, which assumes the entire world of an agent is restricted to one physical space. We describe this setup in more detail in Sec. 3.1. Subsequently, to learn representations under this setup, we provide an overview of our framework, Test-Space Training (`TST`), in Sec. 3.2.

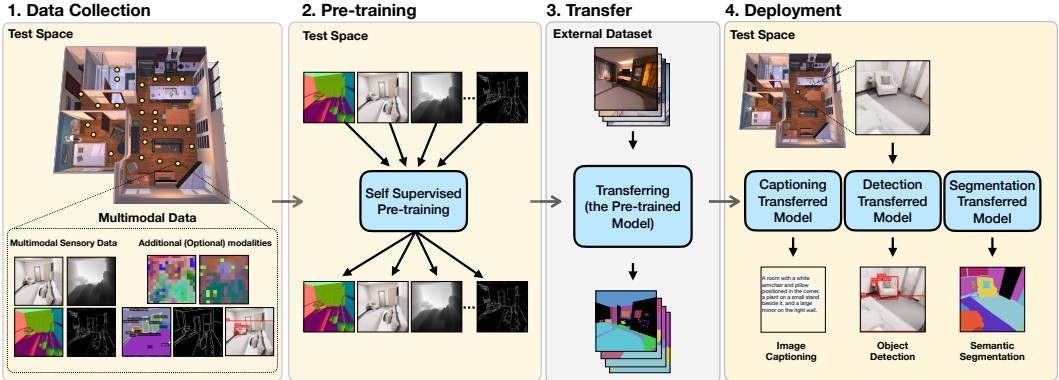

Figure 2: **TST framework. 1)** First, we collect (multimodal) data from the test space (Sec. 3.2.1). **2)** We then use this data for self-supervised multimodal pre-training (Mizrahi et al., 2023; Oquab et al., 2023) (Sec. 3.2.2). **3)** After pre-training, the model is fine-tuned on a small external transfer dataset to solve a desired downstream task, e.g. semantic segmentation (Sec. 3.2.3). **4)** This model is subsequently deployed and evaluated in the same test space where it was pre-trained (Sec. 4).

## 3.1 PROBLEM SETTING

As described in Sec. 1, we construct a sandbox setup to study multimodality as a source of self-supervision in a controlled manner. We do so by restricting the user device to a specific physical space. For instance, a household robot spawned in a building, or as we call it, the test space. It assumes that the pre-training and downstream evaluation happen in the same space. This is a valid assumption, as the pre-training is self-supervised and does not assume any task label information from the test space. This setup allows us to *control* the pre-training and evaluation data distribution and perform various analyses (Sec. 4.4, Sec. 4.5) to identify how each design factor contributes to the performance of cross-modal learning as a self-supervised learning signal.

This sandbox leads to a **specialization** setup, which requires a model to be performant in a specific user space, as we refer to in this work, *the test space*. This represents *real-world deployment* scenarios for various user devices such as household robots, AR/VR goggles, and digital assistants. These applications often require models to be performant in a given user space, irrespective of their generalization abilities elsewhere. Next, to describe how we can learn a specialized model for a given test space, we discuss our framework Test-Space Training (TST).

## 3.2 TEST-SPACE TRAINING

To learn specialized representations for a given space, we propose Test-Space Training (TST). We present an overview of our framework in Fig. 2. It starts by collecting unsupervised, multimodal pre-training data in the test space. Concretely, we assume access to the sensory data sampling function in the test space, denoted as $x \sim p_{\text{space}}(x)$, and use it to collect a pre-training dataset $D_{PT} = \{x_i\}$ (Sec. 3.2.1). Besides RGB images, we also leverage other sensors available on the device, e.g., depth and surface normals. In real-world deployment, this set can be expanded significantly to other common sensors, such as IMU, microphone, radar, and occasionally haptics. We use this data to pre-train a self-supervised model $f : X \rightarrow h$ that maps RGB images into representations (Sec. 3.2.2). We evaluate this model on downstream tasks via transfer learning (Sec. 3.2.3).

### 3.2.1 MULTIMODAL DATA COLLECTION

As noted in Sec. 3.2, we assume access to the sensory data sampling function in the test space, denoted as $x \sim p_{\text{space}}(x)$ to collect pre-training data, $D_{PT}$. This can represent capturing data at various vantage points (Eftekhar et al., 2021), or a video walkthrough (Baruch et al., 2021; Yeshwanth et al., 2023) to cover the test space. In addition to RGB frames, we also collect data from various sensors and modalities available on the user device being deployed in the test space. Additionally, we can also process this data to create more optional modalities as illustrated in Fig. 2. As we later show

in Sec. 4.4 and Fig. 4, scaling this rich set of modalities in the multimodal, test-space data is more effective than scaling to diverse unimodal data from external sources. It is also worth noting that such a dataset of potentially repetitive images from the same space is related to findings in developmental psychology research suggesting that infants observe highly redundant visual data (Jayaraman et al., 2015; Slone et al., 2019). We defer more implementation details for our data collection to Sec. 4.1. This stage results in a multimodal sensory dataset, which we use for self-supervised pre-training.

### 3.2.2 Self-Supervised Pre-training

We employ self-supervised learning to pre-train a model $f$ on the multimodal data $D_{PT}$ collected in the test space. Akin to standard self-supervised pre-training, this model learns task-agnostic representations that are useful for various downstream tasks. As the objective of this work is to study the role of multimodality as supervision, we leverage cross-modal learning as our primary pre-training objective. We implement it using multimodal masked modelling (Bachmann et al., 2022), and refer to the resulting `TST` variant as `TST-MM`. We provide more implementation details on the model naming notation, architecture, and objective in Sec. 4.1.

To study various aspects of cross-modal learning, we begin with the choice of modalities in `TST-MM`. This is a critical choice, as it directly drives the quality of representation learned by the model. We explore a minimal setup with only hardware-based modalities[3] in Sec. 4.2, and scale them further to leverage internet-based pseudolabel networks as additional modalities in Sec. 4.3. We find that `TST-MM`, with the scaled set of modalities, achieves state-of-the-art performance in the test space (Tab. 1). We also show various other analyses on the role of multimodality (Sec. 4.4) and pre-training data source (Sec. 4.5) in specialization.

It is worth noting that although the objective of this work is to study cross-modal learning, `TST` can also support other self-supervised objectives. We explore unimodal learning (RGB-only) objectives (He et al., 2021; Oquab et al., 2023), and find that multimodal learning is the most performant in this setup, consistently achieving superior performance (App. H).

### 3.2.3 Evaluation via Transfer Learning

We evaluate the effectiveness of the pre-trained model $f$ using transfer learning. We add a task-specific head $g$ and finetune the resulting model $g \circ f$ on various downstream tasks, following standard practice in self-supervised learning (Mizrahi et al., 2023; He et al., 2021). For this, we consider a small transfer dataset $D_t$ with task-specific annotations, collected in an external space, disconnected from the test space. Importantly, *we do not have access to any task-specific annotations from the test space itself*, i.e., $D_t$ and $D_{PT}$ are sampled from different distributions. We benchmark against several off-the-shelf vision models (Radford et al., 2021; Oquab et al., 2023; Bachmann et al., 2024), by finetuning them on the transfer data, $D_t$, as discussed in Sec. 4.3.

## 4 Experiments

We present the results as follows. Sec. 4.1 describes our experimental setup and baselines. Sec. 4.2 starts with a minimal setup, which only assumes access to hardware-based modalities. This setup is fairly performant, but leaves room for further improvement compared to internet-based generalists. Therefore, in Sec. 4.3, we extend the modality dictionary by leveraging off-the-shelf pseudolabel networks, and find that this leads to state-of-the-art models for the test space (Tab. 1).

Next, we leverage our sandbox to draw insights on the role of multimodality (Sec. 4.4) in building specialized models with `TST`. Lastly, Sec. 4.5 explores the role of pre-training data from the test space in achieving specialization.

### 4.1 Experimental Setup

**Datasets.** We experiment using three datasets:
*1. Scannet++* (Yeshwanth et al., 2023) is a large-scale dataset of ***real-world*** indoor spaces containing sub-millimeter resolution scans, paired with DSLR and iPhone RGB images. We use 8 scenes as the

---

[3]we describe the setup in more detail in Sec. 4.1

test space, and use a mix of iPhone and DSLR images from these scenes for pre-training.
*2. Replica* (Straub et al., 2019) provides high-quality 3D reconstructions of *real* indoor spaces. We use 5 scenes as the test space, and use rendered RGB-D images for pre-training.
*3. ProcTHOR* (Deitke et al., 2022) includes procedurally generated house-like environments. We use 5 procedurally generated houses as the test space, unless specified otherwise.

**Pre-training.** For training `TST-MM` models on the dataset $D_{PT}$ collected from a test space, we leverage multimodal masked modelling (Bachmann et al., 2024) as described in Sec. 3.2.2, and train an encoder-decoder transformer model. We use modality-specific tokenizers (Bachmann et al., 2024) to convert all modalities into tokens. We train models across two encoder sizes, ViT-S and ViT-B, which have 8 and 12 encoder layers, respectively. Additionally, we found that mixing RGB images from the transfer was beneficial in pre-training. Please see [App.](#) L for an ablation on this choice. Note that *we do not use any task labels from the transfer set during pre-training, making this stage task-agnostic.* For results in Sec. 4.3, we initialize the model *from scratch*, whereas for adaptation (Sec. 4.3, Adaptation through `TST`) we initialize from 4M-21 (Bachmann et al., 2024).

**Notations.** We refer to `TST` variants with different objectives as `TST-MM` for multimodal masked modeling, `TST-MAE` for unimodal masked modeling, `TST-DINO` for DINOv2 (Oquab et al., 2023). Unless specified otherwise, we refer to the multimodal version (`TST-MM`) as `TST`.

**Transfer and Evaluation.** For all datasets, we use an external set of scenes that are different from the test space to collect a small transfer set ($D_t$) with task-specific annotations. We evaluate the transferred models in the test space on semantic segmentation (Scannet++, Replica, ProcTHOR), object detection (Scannet++, ProcTHOR) and image captioning (ProcTHOR). We provide more details on the transfer and evaluation setup in the [Appendix.](#)

**Modalities.** For Scannet++ (Yeshwanth et al., 2023), we use RGB images captured by DSLR and iPhone cameras. For Replica (Straub et al., 2019) and ProcTHOR (Deitke et al., 2022), we render *RGB-D* from the test space using onboard sensors. We also include *Surface normals* and *Canny Edges* as they can be extracted from RGB and depth respectively using simple operations. We refer to these 4 modalities (RGB, Depth, Surface normals, and Canny Edges) as *sensory* in Sec. 4.2 and thereafter. In Sec. 4.3, we discuss how we can further scale the number of modalities using off-the-shelf networks.

**Baselines.** We compare against several baselines:
• *Scratch - no pre-training.* We compare against both unimodal scratch, which takes only RGB images as input, and multimodal scratch, which inputs all modalities available to `TST-MM` during transfer training and evaluation. The latter baseline indicates that the performance is not owed to merely having multiple modalities, but rather performing cross-modal pre-training.
• *Large-scale generalist pre-training baselines.* We evaluate 4M-21 (Bachmann et al., 2024), DINOv2 (Oquab et al., 2023), and CLIP (Radford et al., 2021) as recent strong generalist (self-supervised) baselines, trained on large-scale datasets via unimodal and multimodal learning. To ensure fair comparison, we finetune these models, with the same transfer dataset ($D_t$) as `TST`.
• *Task specialist baselines.* We perform evaluations using Mask2Former (Cheng et al.), ViTDet (Li et al., 2022), SAM (Kirillov et al., 2023), and LLaVA-1.5 (Liu et al., 2023) as established task-specific baselines for semantic segmentation, object detection, and image captioning. Similar to generalist baselines, we finetune these models, with the same transfer dataset, $D_t$, that we use for `TST`.

We finetune the baselines with the same transfer data $D_t$, and use only RGB images as input for transfer and evaluation (except for the multimodal scratch baseline). An extensive hyperparameter search was done for fair comparison. We provide more details on our experimental setup, including pre-training and transfer training hyperparameters in [App.](#) W.

## 4.2 How far can we go with no external access?

A central design choice in our setup is the choice of modalities to perform cross-modal learning over. It directly controls what representation is learned by the model. We begin by exploring a setup that starts with a set of sensory modalities (as described in Sec. 4.1) that are easily acquired via hardware sensors. This is a minimal setup, which, when combined with cross-modal learning, allows bootstrapping a vision representation for the test space, with *no external access*. Fig. 3 shows, on Scannet++ (Yeshwanth et al., 2023), that pre-training with just multimodal sensory data from the test space can perform competitively with generalist models such as DINOv2 (Oquab et al., 2023),

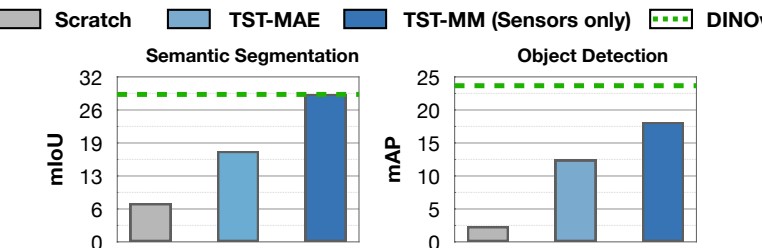

Figure 3: **How far can we go with no external access?** We compare results of pre-training using large-scale Internet data (DINOv2 (Oquab et al., 2023) on 142M images) with using only data collected from a test space with onboard sensors, TST-MM (Sensors). We show segmentation and detection results on a test space from the Scannet++. *We find that, with no external access, TST-MM with sensory modalities, and just multimodal data from the test space, is competitive with large-scale Internet-based pre-training.*

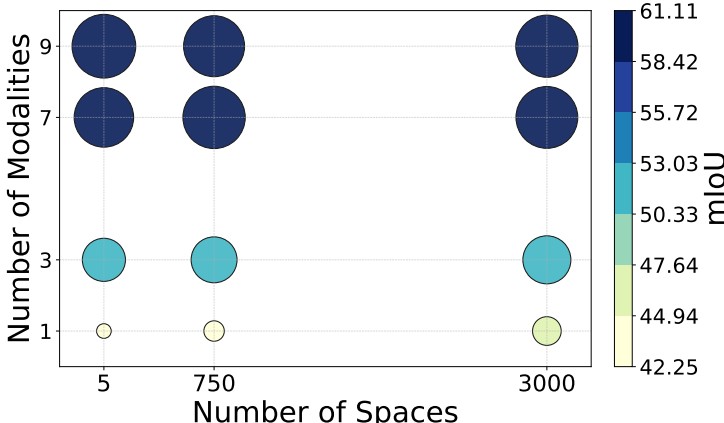

Figure 4: **Modality scaling vs data scaling.** We study the tradeoff between collecting unimodal pre-training data from more spaces versus scaling modalities in the test space (here, 5 houses). The size of each circle is proportional to the mIoU performance on segmentation. We find that scaling the number of modalities within the test space yields better performance than scaling data by including external spaces. All models use the ViT-S backbone.

pre-trained on large-scale external (internet-based) data. Additionally, note that although unimodal pre-training (TST-MAE) improves over learning from scratch, multimodality with TST-MM (Sensors only) performs the best.

This suggests that with cross-modal learning on just sensory modalities, *we can build highly performant models for the test space, without any external access*, and achieve competitive performance with large-scale Internet-based pre-training (Oquab et al., 2023). However, it is worth noting that the gap in performance, with state-of-the-art generalists, suggests that this set of modalities, with the current multimodal architecture and objective, is sub-optimal for the semantic downstream tasks evaluated here.

## 4.3 SCALING MODALITIES IN THE TEST SPACE

Sec. 4.2 discusses that although TST with cross-modal learning with only sensory modalities achieves competitive performance against internet-based generalists (Oquab et al., 2023), it is not performant enough to replace it yet. Therefore, to enrich this representation further, we draw inspiration from recent progress in multimodal foundation models (Bachmann et al., 2023), and scale up the set of

---

[2]Task-specific methods used for each result, in order: SAM (Kirillov et al., 2023) (segmentation), ViTDet (Li et al., 2022) (detection), and LLaVA (Liu et al., 2023) (captioning)

| | Method | Semantic Segmentation | | | Object Detection | | Captioning | |
|---|---|---|---|---|---|---|---|---|
| | | Scannet++ mIoU | ProcTHOR mIoU | Replica mIoU | Scannet++ mAP | ProcTHOR mAP | ProcTHOR CIDEr | SPICE |
| No Pre-training | Unimodal Scratch - no pre-training | 7.49 | 28.62 | 9.23 | 2.35 | 24.59 | 17.1 | 14.8 |
| | Multimodal Scratch - no pre-training | 7.82 | 26.29 | 10.03 | 3.76 | 19.19 | 11.0 | 10.5 |
| Generalist Pre-training | 4M (RGB-only) / MAE (He et al., 2021) | 13.74 | 46.29 | 18.18 | 18.31 | 37.17 | 30.4 | 19.1 |
| | 4M-21 (Bachmann et al., 2024) | 27.59 | 53.24 | 26.30 | 25.91 | 41.43 | 36.2 | 20.3 |
| | DINOv2 (Oquab et al., 2023) | 30.60 | 54.50 | 26.72 | 23.67 | 40.28 | 14.7 | 13.5 |
| | CLIP (Radford et al., 2021) | 23.19 | 48.66 | 20.92 | 19.75 | 38.47 | 18.4 | 16.2 |
| Task Specialist | Task Specific Methods [2] | 34.75 | 56.72 | 28.51 | 23.59 | 44.10 | **40.60** | **21.00** |
| Specialist Pre-training | TST-MM | 34.49 | **60.85** | 32.87 | 31.54 | 49.38 | 34.3 | 20.4 |
| | TST-MM (adapted) | **36.44** | 60.59 | **34.53** | **35.83** | **51.25** | 39.9 | 20.5 |

Table 1: **Multimodal Test-Space Training (`TST-MM`) outperforms both strong generalists and task specialists across tasks.** We evaluate semantic segmentation, object detection, and image captioning. All models use ViT-B backbones, except SAM (Kirillov et al., 2023) (ViT-H). `TST-MM` (adapted) refers to fine-tuning 4M-21 on test-space data. On segmentation and detection, `TST-MM` consistently outperforms Internet-based generalists and matches or surpasses specialists. On captioning, `TST-MM` (from scratch), despite no text pre-training, matches 4M-21 trained on CC12M image-text pairs; `TST-MM` (adapted) surpasses 4M-21 and approaches LLaVA-1.5 (Liu et al., 2023).

modalities using off-the-shelf pseudolabel networks. Next, we compare `TST` with this scaled set of modalities (`TST-MM`), against Internet-based generalists and task specialist models. Lastly, we describe how `TST` can also enable adapting a pre-trained generalist to the test space.

**Additional Modalities.** We create new modalities by pseudolabeling the collected RGB frames. We use neural network feature maps (Radford et al., 2021; Girdhar et al., 2023), *SAM edges* (Kirillov et al., 2023), *bounding boxes* from ViTDet (Li et al., 2022), and *semantic segmentation masks* from Mask2Former (Cheng et al.). For a fair comparison, we also include these pseudolabeling networks as baselines and show that `TST-MM`, trained *from scratch*, outperforms all of them (see Tab. 1, and App. N). Note that `TST-MM`, which leverages these additional modalities, no longer qualifies for *no external access*, as it is akin to distilling the off-the-shelf pseudolabeling networks which were trained on large-scale external data. However, note that this distillation happens only on test-space data, therefore alleviating the need to directly access external data by leveraging off-the-model outputs as modalities. This is reminiscent of recent findings on attention transfer (Li et al., 2024a), which show that only distilling the attention patterns from a pre-trained teacher performs similarly to full finetuning of the model.

**`TST` vs. generalists.** Tab. 1 shows quantitative results for `TST-MM`. We compare against and outperform generalist models (MAE, DINOv2, 4M-21, and CLIP) trained on large-scale Internet datasets. *This suggests that we can outperform generalist models by using cross-modal learning on multimodal data from the test space.* Figure 10 shows qualitative improvements of `TST` over generalist Internet pre-training.

**`TST` vs. task specialists.** We also show that `TST-MM` outperforms or is on par with off-the-shelf task specialist models on semantic segmentation (Kirillov et al., 2023) and object detection (Li et al., 2022). For image captioning, despite not seeing any text data during pre-training, `TST-MM` performs on par with 4M-21 (Bachmann et al., 2024) that was pre-trained on large-scale image-text data (Changpinyo et al., 2021), showing the effectiveness of the learned representation.

**Adaptation through `TST`.** Tab. 1 also presents results when `TST-MM` adapts an existing generalist model to the test space. Unlike all `TST-MM` models discussed above that start from scratch, we start from a pre-trained 4M-21 model and fine-tune it on data from the test space, using multimodal masked modeling. The resulting model, `TST-MM` (adapted), significantly improves over 4M-21 (Bachmann et al., 2024) in the test space. Therefore, *`TST` can also serve as an adaptation mechanism for Internet-based models, making them more performant in the test space for downstream tasks.*

## 4.4 MULTIMODALITY IN TST

We show that leveraging cross-modal learning, and specifically, multimodal masked modelling, is a key component in enabling performant models in the test space. In this section, we leverage the same sandbox setup (Sec. 3.1) to perform controlled analysis on various aspects of multimodality in `TST`.

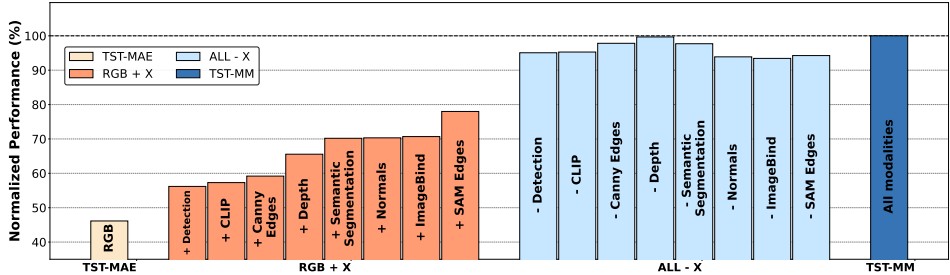

Figure 5: **Contribution of different modalities to `TST` performance.** We study the effect of each modality on `TST` by dropping one combination from `TST-MM`, and adding one to `TST-MAE` (RGB-only `TST`). We use the ViT-S backbone. We find that even though some modalities provide higher gains than others when added to the RGB-only `TST-MAE`, the performance of `TST-MM` stays relatively stable, agnostic to the choice of the dropped modality. *This indicates that no single modality is responsible for `TST-MM`'s performance, but rather their collective interplay, i.e., multimodality.*

**Can we substitute large-scale data with more modalities?** We study the trade-off between using *smaller-scale but modality-rich test-space data*, versus *large-scale unimodal external data* (RGB-only). We use ProcTHOR (Deitke et al., 2022) to generate similar spaces (IID with the test space) and leverage them as an external data source. Starting from unimodal pre-training in the test space, Fig. 4 shows that scaling the number of modalities in the test space yields significantly better performance than increasing the amount of unimodal data from external sources. Additionally, we find that although scaling unimodal external data improves over the unimodal baseline in the test space. However, for a higher number of modalities, for a ViT-S backbone, we find that the performance is relatively stable when external data is added. This suggests: *For building high-performing models in a specific test space, collecting data within that space using a richer set of modalities is often more effective than relying on large-scale, unimodal data collected from external sources.*

**Is the choice of modalities important for the effectiveness of the multimodal pre-training?** We investigate whether all modalities contribute similarly to multimodal pre-training, as shown in Sec. 4.2 and 4.3, or if there is a single modality that contributes the most. We present two ablations in Fig. 5. First, we examine all pairs of two modalities starting with RGB, i.e., all {RGB, $X$} combinations. Adding any modality improves performance, with some showing greater benefits than others (e.g., *SAM edges* increase performance by an absolute 7.8%), but none matches the performance of using all modalities. Second, we examine all sets of eight modalities by removing one modality from `TST-MM`, except RGB (which remains as the input during finetuning and evaluation). We find low variance between different sets, indicating that no single modality is irreplaceable and that other modalities can compensate for the absence of useful ones[4]. For example, removing the *SAM edges* modality reduces results by only 1.5%, compared to its absolute 7.8% improvement when added to RGB alone. *Thus, once a large set of performant modalities is collected, the performance is relatively stable to the exact combination of them, reducing the need to engineer an optimal set.*

**How does the performance of `TST-MM` scale with modalities?** Fig. 6 shows the performance of `TST-MM` as we increase the number of modalities. Due to the combinatorial complexity of studying all possible combinations, we only sample all possible options for two (RGB+X) and eight (All-X) modalities, where here X is the modality added or dropped. For other modality counts, we randomly sample 8 modality sets and report the average performance on the plot[5] *We find that the performance of `TST-MM` scales well with more modalities, agnostic of the exact modality combination, and with decreasing variance between subsets.*

## 4.5 MEASURING SPECIALIZATION WITH TST

As discussed in Sec. 3.1, our sandbox presents a **specialization** setup. We define *specialization* as the measure of how performant a model is, on a downstream task, in a given test space, while

---

[4]Note that the nature of modalities does play a role in this large set. As noted in Sec. 4.2, only using sensory modalities was not sufficient to achieve the most performant models in the test space, and adding pseudolabel modalities (Sec. 4.3) was necessary to improve further.

[5]This is necessary to ensure fairness, as the modality axis is not uniform. For instance, going from modality count 1 to 2 can bring about different gains, depending on the exact modality, as can also be seen in Fig. 5.

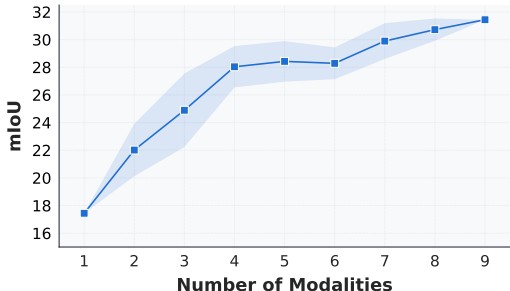

Figure 6: **Scaling the number of modalities for `TST-MM`.** We report the performance of `TST-MM` as we scale the number of modalities. We begin with only the RGB modality and incrementally add more. *Increasing the number of modalities improves performance, while the variance in performance due to any specific modality decreases.*

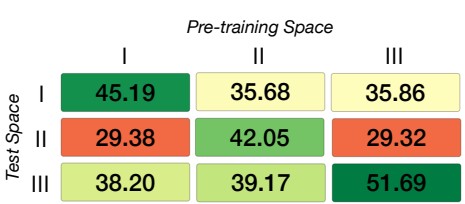

Figure 7: **Do we need the same test space for pre-training and evaluation?** We perform cross-space analysis by pre-training and evaluating performance on different spaces. Each column and row represents a pre-training and test space. *Performance is best along the diagonal, where pre-training and evaluation are in the same space.*

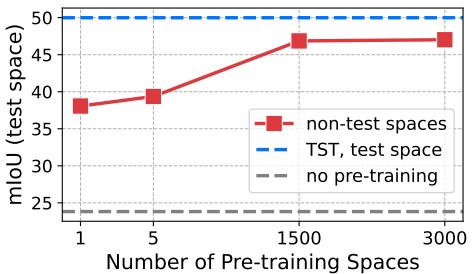

Figure 8: **How many spaces is one test space worth?** We study whether test-space data for pre-training can be substituted with data from similar but nonidentical spaces. We compare performance on the test space between `TST` and models pre-trained with a ViT-S backbone on an increasing number of IID houses. *We find that using as many as 3000 spaces cannot match pre-training in the exact test space*, underscoring the usefulness of test-space specialization with `TST`.

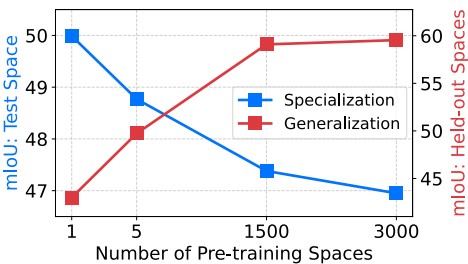

Figure 9: **Specialization-generalization trade-off.** We pre-train ViT-S models on data collected from a growing number of spaces, starting with a single test space and adding data from other IID spaces. The blue curve shows performance in the test space (specialization); the red curve shows performance on 100 held-out IID spaces (generalization). *As we add more pre-training spaces, test-space performance decreases while held-out performance improves, revealing the specialization-generalization trade-off.*

forgoing other spaces. E.g., a model is specialized to space A if it performs well in test space A but is inferior in Space B. This is in contrast to *generalization* in conventional machine learning, which measures performance on a set of held-out spaces. As described in Sec. 3.2.1, `TST` collects pre-training data in a test space to pre-train a model for that space. In this section, we explore, given the same cross-modal learning objective, the role of this pre-training data from the test space itself.

First, we measure it by cross-evaluating models pre-trained on two different test spaces, showing that space-specific pre-training performs the best. Then, we explore if we can substitute data from the test space with data from many (thousands of) similar spaces during pre-training, effectively asking: *how many spaces is the test space worth?* Third, we explore whether a single model can exhibit both specialization and generalization capabilities and show the *specialization-generalization trade-off*.

**`TST` effectively specializes to a test space.** Fig. 7 shows the performance of models pre-trained and evaluated on different test spaces. We find that in all cases pre-training the model in the corresponding test space is the best, demonstrating the practical value of specialization. We observe similar trends for other pre-training objectives (App. P).

**How many spaces is a single test space worth?** If not one space, data from how many similar spaces can substitute test-space data? Similar to Sec. 4.4, we use ProcTHOR (Deitke et al., 2022) to generate a large number of similar houses and pre-train models using an increasing number of them.

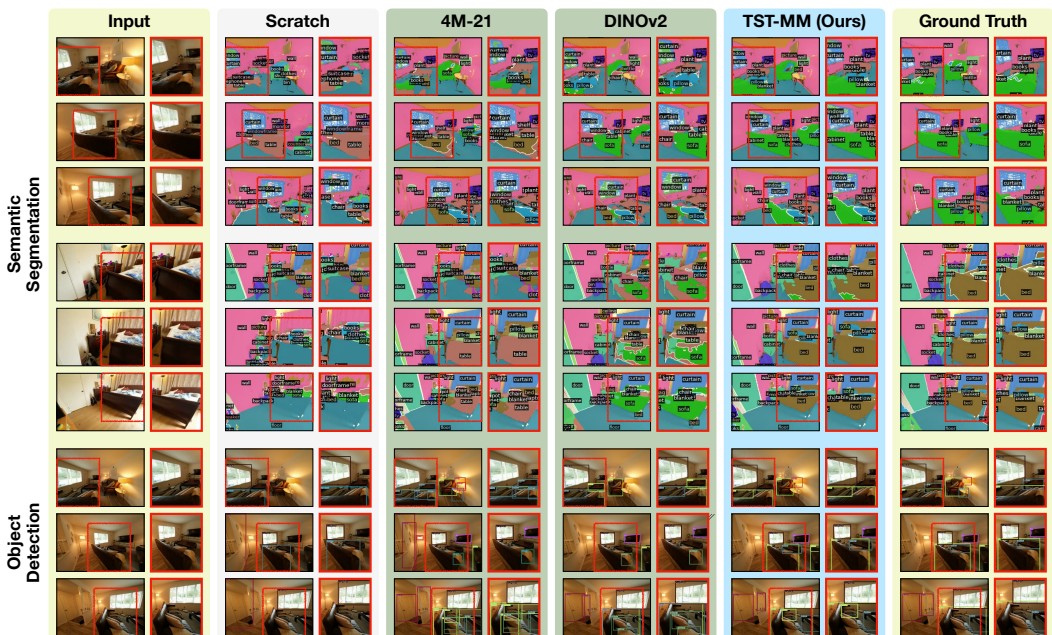

Figure 10: `TST-MM` **predictions across different tasks**. We present qualitative results for `TST-MM` against various baselines, including scratch (no pre-training) and Internet-based pre-training on **real-world scenes** from Scannet++ (Yeshwanth et al., 2023). `TST-MM` predictions are notably more consistent across both tasks. Note how `TST-MM` predicts the same object (magnified in red boxes) more accurately and robustly across various viewpoints, as compared to generalist models like 4M-21 (Bachmann et al., 2024) and DINOv2 (Oquab et al., 2023).

Fig. 8 shows the performance of each model on the test space not seen during pre-training compared to pre-training on the corresponding test space. We find that *even thousands of similar spaces are not enough to substitute pre-training on the exact same space that we deploy in*.

**Specialization-generalization trade-off.** We observed that the best performance on a given test space is achieved when pre-trained on data from the same space. However, we would not expect this specialized model to generalize well to new houses. Can we keep (or improve) this specialization performance while gaining generalization capabilities by adding more houses during pre-training in addition to the test house? We perform an analysis, where we fix the model and dataset size, thereby fixing the total compute spent, and study the effect of trading off test-space data, with external data from similar houses. Fig. 9 shows that as we add more houses during pre-training, the performance on the held-out new houses increases, as expected. However, the performance on the original test space drops compared to the specialist single-space pre-training, demonstrating a *specialization-generalization trade-off* of the pre-trained model.

**Additional results in Appendix.** Besides the analysis presented here, in the Appendix, we present more experiments on deploying `TST` in the wild (App. F), `TST` using other self-supervised objectives (App. P), `TST` as label propagation (App. C), the role of the transfer dataset mix-in during pre-training (App. L), results for cross-modal retrieval (App. E), and qualitative videos on real-world spaces on our website. Additionally, along with our code, we open-source a Swift-based iOS application (App. R) that leverages the ARKit API[6] to stream the outputs of various sensors, including RGB, LiDAR, IMU, magnetometer, and ambient lighting.

## 4.6 BROADER IMPACT OF TST

Our work sheds light on the critical role of data and data sources in building vision systems. The setting studied in `TST` suggests that it is possible to achieve competitive results without relying on large, diverse internet-based datasets Changpinyo et al. (2021); Schuhmann et al. (2022) that

---

[6]https://developer.apple.com/documentation/arkit

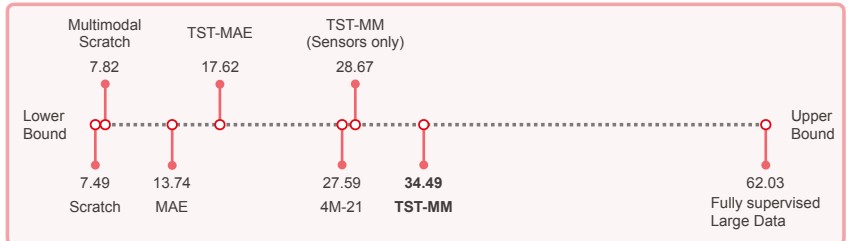

Figure 11: **Quantitative summary of TST**. The lower bound is scratch (i.e., no pre-training and learning using the external transfer set only). The upper bound is approximated by a fully supervised model trained with a large number of annotated segmentation images. The gap between the lower and upper bounds is the playfield for the pre-training methods to fill. All methods share the same model architecture. The results here correspond to semantic segmentation on Scannet++ (Yeshwanth et al., 2023).

essentially require the data of different users to be harvested and mixed. It shows that training on only the deployment space data is an alternative worth considering and investigates the requirements of making that viable (e.g., utilizing multimodality being critical for achieving good results). This paradigm enables segregating the data of different users, which can enable a complete in-house, local training setup under full user control for privacy-critical scenarios, thereby avoiding any potential data contact with the external world. We provide an extended discussion on this in App. V.

## 5 DISCUSSION AND FUTURE WORK

Our work presents a controlled study for leveraging multimodality as self-supervision for specialization. We conduct this study in a sandbox, which constrains the operating space of a multimodal device to a specific test space. It provides controllability over pre-training and evaluation data distributions, resembles developmental settings of biological agents (Pereira et al., 2014), and mirrors real-world deployment conditions (Sec. 3.1). To learn representations in this sandbox, we present Test-Space Training (TST), a framework that performs unsupervised, multimodal pre-training data collection in the test space, and then pre-trains a model on it using cross-modal learning.

We present a consolidated picture of our results in Fig. 11 and highlight the following key takeaways:

• **TST-MM yields the most performant model in the test space.** It currently sits halfway between the bounds. It leverages cross-modal learning with sensory and off-the-shelf pseudolabelled modalities, with just test space data for pre-training. Note that this model does not qualify for claiming "no external access" (Sec. 4.3).

• **Distilling on test-space data vs external data.** The distillation of pseudo-labeler network outputs as modalities (Sec. 4.3) is more effective on test-space data than on external data. Comparing TST-MM with 4M-21 (Bachmann et al., 2024) shows that, as the two are equivalent in nearly all aspects, except that 4M-21 distills the pseudo-labelers on external data (CC12M) while TST-MM distills the same pseudo-labelers only on test-space data. The same observation was consistently made for various settings in Fig. 23 of the App. Q.

• **How far can we go with no external access?** As discussed in Sec. 4.2, the TST-MM (Sensors only) uses no external data, directly or indirectly, as it only uses on-device sensors as modalities. It covers roughly 1/3 of the way, providing a nontrivial value, and is competitive with internet-based generalist, 4M-21, trained on large-scale internet data (CC12M (Changpinyo et al., 2021)).

• **Multimodality as supervision is effective.** Generally, multimodality is useful as the multimodal models consistently outperform their single-modal counterparts (e.g., TST-MM vs TST-MAE or 4M-21 (Bachmann et al., 2024) vs MAE (He et al., 2021)). Also, note that multimodality was necessary to make TST-MM a viable alternative to internet-based generalists (Oquab et al., 2023; Radford et al., 2021) in the test space, as TST-MAE performs subpar in comparison.

Our work serves as a problem setup, rather than a complete solution for studying specialization via leveraging deployment space data in a controlled manner. We find that although cross-modal learning

with multimodal masked modelling achieves state-of-the-art performance in the test space, there is still room for improvement (Fig. 11) to close the gap with the upper bound.

It also presents an interesting opportunity to study the role of structure against scale (d'Ascoli et al., 2021) in learning vision representations. Internet-based generalists leverage scaling across model and dataset size (Dosovitskiy et al., 2021; Oquab et al., 2023; Radford et al., 2021) to achieve broad generalization abilities. On the other hand, `TST` forgoes scaling to diverse data by adding more structure to the test space data itself. In this paper, we explore this structure via leveraging multimodality as a source of self-supervision and explore its tradeoff to scaling unimodal external data (Fig. 4).

In future research, we are interested in advancing the current pre-training methodology to inch closer to the upper bound by incorporating other inductive biases such as multi-view consistency (Luo et al., 2020). Additionally, we are interested in exploring various hardware-based modalities such as IMU, gyroscope, and magnetometer.

**Acknowledgements.** This work was supported under project ID **43** as part of the Swiss AI Initiative, through a grant from the ETH Domain and computational resources provided by the Swiss National Supercomputing Centre (CSCS) under the Alps infrastructure. This material is based on work that is partially funded by an unrestricted gift from Google. This work has received funding from the Swiss State Secretariat for Education, Research and Innovation (SERI). We also thank Daniel Filipe Jana and the EPFL SCITAS team for their support. The authors also thank Chandan Yeshwanath for their help with the Scannet++ (Yeshwanth et al., 2023) dataset, and Roman Bachmann for useful discussions.

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

# APPENDIX

## A   ADDITIONAL RELATED WORK.

**Domain Adaptation** in vision (Li et al., 2017; Zhou et al., 2021) addresses the gap between a source domain, where abundant data is available, and the target domain, where limited (Shu et al., 2019; Liu et al., 2024) or no data (Dong et al., 2021; Ganin & Lempitsky, 2014) are available. TST, when initialized from an Internet-based model, as presented in Tab. 1, can be seen as an instantiation of adapting a generalist model to the test space. However, TST differs by learning task-agnostic representations by self-supervised pre-training in the test space, as opposed to domain adaptation, which generally adapts a pre-trained task-specific network (Xu et al., 2021; Kang et al., 2019).

**Semi-Supervised Learning** refers to a line of work that attempts to learn a task from a limited labeled dataset and massive unlabeled data (van Engelen & Hoos, 2019). Clearly, it involves consistency regularization (Berthelot et al., 2019; Sohn et al., 2020; Xie et al., 2019) and pseudo-labeling  (Guo et al., 2022; Chen et al., 2021; Zhang et al., 2021; Liu et al., 2021a) to generate supervision of unlabeled data, followed by joint training. Our framework, TST is closer in spirit to *self-supervised learning*, as it tries to learn a task-agnostic representation for the test space, that we transfer for various downstream tasks like segmentation, detection and image captioning. Under semi-supervised learning, specialization with TST can be posed as using unlabelled data from the test space, as opposed to other sources like Internet or similar spaces.

**Embodied Active learning.** In another line of work, SEAL (Chaplot et al., 2021), Interactron (Kotar & Mottaghi, 2022) learn a reinforcement learning-based policy to collect supervision in a house to finetune an off-the-shelf MaskRCNN (He et al., 2017), or observe additional frames for multi-frame inference for object detection. As opposed to focusing on adapting task-specific models, we focus on learning task-agnostic pre-trained representations over a test space.

## B   SPECIALIZATION CAN REDUCE DEPLOYMENT COST

We find that cross-modal learning on test-space data yields specialist models that can outperform internet-pretrained generalist models (Tab. 1). This specialization, however, introduces an additional cost: unlike generalist pre-training, which is performed once and reused across deployment spaces, TST requires pre-training a separate model for each target space, causing training compute to scale with the number of deployment spaces.



However, TST can also shift the efficiency tradeoff. As shown in Fig. 12, TST achieves strong performance with a smaller transformer backbone, partially offsetting the cost of deployment-specific pre-training through reduced inference compute. In particular, because ViT-S is $4\times$ faster at inference than ViT-B, the one-time cost of specialist pre-training can be amortized over repeated deployment: after a break-even point, serving a smaller specialist model becomes cheaper than serving a larger generalist model. This observation is consistent with re-

**Figure 12: TST can achieve similar performance with smaller model sizes.** We find that a specialized model with a ViT-S backbone can outperform internet-based generalist models such as DINOv2 and 4M-21 with a ViT-B backbone, which is 4x more computationally expensive to run inference on.

cent work on specialist pre-training for language models (Baek et al., 2026), which finds that smaller specialist models can be both cheaper and more performant to serve than large-scale generalist models.

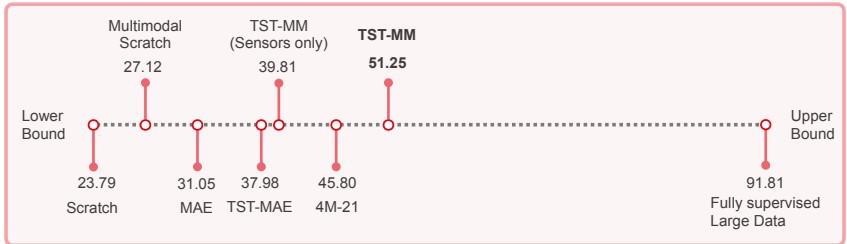

Figure 13: **Label Propagation with `TST`**. We find that the trends in performance under the Label Propagation, with no transfer set distribution shift, are similar to what we observe in Tab. 1 and Fig. 11, thereby reassuring the efficacy of pre-training in the test space. The lower bound here is learning from scratch (i.e., no pre-training and learning using the small transfer set only). The upper bound is approximated by a fully supervised model trained with a large number of annotated segmentation images from the test space. All methods share the same model architecture. The results here correspond to semantic segmentation on Scannet++ (Yeshwanth et al., 2023).

## C   LABEL PROPAGATION WITH TST

As mentioned in Sec. 3.2.3, all results presented so far use external transfer images — i.e., the annotated images used for training the transfers are from other spaces, rather than the test space. This choice matches a real-world setup, as it is infeasible for a user to spend time annotating images from their space. However, this introduces a minor suboptimality in quantifications, since the transfer set may exhibit distribution shifts from the test space (e.g., some test objects may not appear in the transfer set or have vastly different appearances). This mismatch makes it difficult to attribute poor results primarily to ineffective pre-training.

To study the effect of pre-training, we experiment with a controlled setup where transfer images are a small set ($\approx 20$) from the same test space. This is akin to having a few annotated images from the test space and expecting the model to propagate the labels to the rest of the images of the test space. Therefore, we name this setup as *Label Propagation*. This allows us to leverage this setup as a sandbox for studying various self-supervised pre-training methods in the test space. We present the results of various TST variants and relevant baselines under this setup in Fig. 13.

## D   DATASET DETAILS.

**1. Scannet++** (Yeshwanth et al., 2023) is a large dataset of real-world indoor spaces containing sub-millimeter resolution laser scans, paired with DSLR and iPhone RGB images.

- **Pre-training dataset.** We use 8 Scannet++ (Yeshwanth et al., 2023) scenes as our test space. We use a mix of iPhone and DSLR images for pre-training, with the iPhone containing 19165 samples and the DSLR dataset containing 15000 samples.
- **Transfer dataset.** We use non-test space buildings for creating a transfer set of 40000 RGB, segmentation pairs. Note that Scannet++ (Yeshwanth et al., 2023) only provides 3D instance annotations, which we project to 2D to create a semantic segmentation dataset.
- **Evaluation.** We evaluate on semantic segmentation in the test space. The test dataset for evaluation contains 3000 RGB image samples. Note that we collect a separate held out set from the test space for this stage.

**2. Replica** (Straub et al., 2019) provides high quality 3D reconstructions of real indoor spaces.

- **Pre-training dataset.** We use Omnidata (Eftekhar et al., 2021), to densely sample Replica meshes corresponding to the 5 scenes to build our pre-training dataset, $D_{PT}$, containing 84889 samples. We defer the details of the sampling procedure to Omnidata (Eftekhar et al., 2021).
- **Transfer dataset.** Similar to Scannet++ (Yeshwanth et al., 2023), we collect a transfer set from another set of Replica scenes that are different than the scenes used during pre-training.

We collect 20000 RGB images and semantic segmentation masks, and use it as our transfer dataset, $D_t$.

- **Evaluation.** We evaluate on semantic segmentation in the test space. We collect a test set of 5000 images and semantic segmentation annotations from the same test space we pre-train on, and report performance on it. We leverage Omnidata annotation pipeline to extract the segmentation labels.

**3. ProcTHOR** (Deitke et al., 2022) It includes procedurally generated house-like environments. We use 5 procedurally generated houses as our test space.

- **Pre-training dataset.** We randomly sample various agent $x, y, z$ positions and orientations along its axis in the test space, and collect RGB-D images at these points. This sampling process yields a total of 163767 samples. We collect data by sampling densely across the test space and use it as our pre-training dataset $D_{PT}$.

- **Transfer dataset.** For the transfer data $D_t$, we collect a small dataset of 20000 RGB and task annotation pairs, from 800 houses generation using a different asset and layout distribution than the pre-training test space, thereby making them out-of-distribution to it.

- **Evaluation.** We evaluate TST and present results on three tasks, namely semantic segmentation, object detection and image captioning. We collect a test set with 5000 samples from the same test space, where we performed pre-training, and report performance on it. We use the AI2-THOR (Kolve et al., 2017) metadata to extract semantic segmentation and object detection labels for evaluations. For captioning, we generate ground truth captions by prompting GPT-4o (OpenAI, 2024) with privileged information, e.g. class names and bounding boxes. Finally, we additionally evaluate our model on cross-modal retrieval (in Sec. E).

# E  ADDITIONAL DOWNSTREAM EVALUATIONS: ZERO-SHOT CROSS-MODAL RETRIEVAL TASK

| Method | Image to Depth | | | Depth to Image | | |
|---|---|---|---|---|---|---|
| | R@1 | R@5 | R@10 | R@1 | R@5 | R@10 |
| 4M-21 (Bachmann et al., 2024) | 1.06 | 2.18 | 3.08 | 1.0 | 2.76 | 3.66 |
| TST-MM | **25.48** | **37.00** | **41.58** | **24.32** | **36.46** | **40.82** |

Table 2: **Zero-shot Cross-modal retrieval.** When performing the image-to-depth and depth-to-image cross-modal retrievals on the test space data using the predicted CLIP embeddings, we observe that the TST-MM method constantly outperforms the Internet-based 4M-21 (Bachmann et al., 2024).

As mentioned in Section 4.1, we present results on zero-shot cross-modal retrieval to further support our framework TST. Specifically, we evaluate the performance of models pre-trained with TST-MM on RGB-to-Depth and Depth-to-RGB retrieval. To perform retrieval using an Internet-based model, 4M-21 (Bachmann et al., 2024) and TST-MM, we utilize their cross-modal generation capabilities by transforming depth and RGB images into CLIP embeddings, and then apply retrieval directly on the CLIP embeddings. Since 4M-21 (Bachmann et al., 2024) and TST-MM generate feature maps for CLIP as the target modality from RGB and Depth images, we apply mean-pooling on the feature maps to obtain global CLIP embeddings. Cross-Modal retrieval evaluates TST-MM on two fronts: i) How well test-space paired modality inputs are aligned in the model representations internally, and ii) How effectively TST-MM can perform cross-modal generalization. For the evaluation, we report zero-shot recall at various thresholds on a test set of 5000 samples from ProcTHOR (Deitke et al., 2022) test space. The results are presented in Tab. 2. We also present qualitative examples in Fig. 14. Note that given our method TST-MM has access to the test space, it can retrieve RGB to Depth and Depth to RGB much more effectively than models based on external data like the Internet.

We find that TST-MM substantially outperforms 4M-21 (Bachmann et al., 2024). The recall performance of TST-MM further increases when evaluated on R@5 and R@10, whereas Internet-based

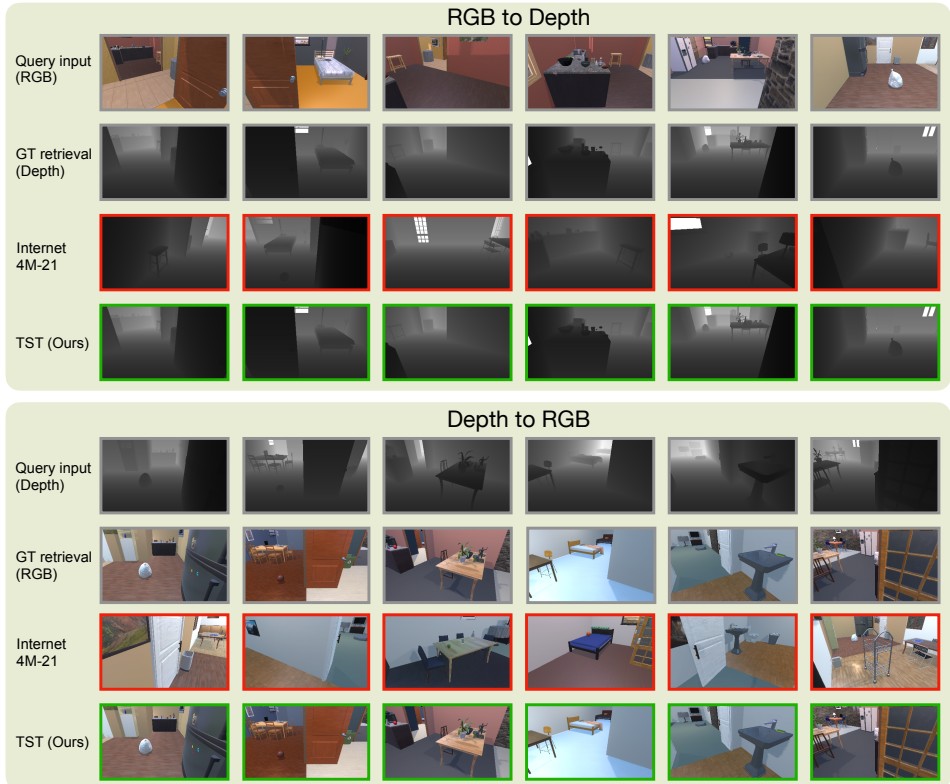

Figure 14: **`TST-MM` cross-modal retrieval predictions**. `TST-MM` retrieves corresponding RGB images from query Depth input and Depth images from RGB input more accurately than the Internet-based 4M-21(Bachmann et al., 2024) model.

4M-21 (Bachmann et al., 2024) shows diminishing returns. This underscores the effectiveness of test-space training, where specialization itself is crucial for learning test-space-aligned representations.

## F `TST-MM` DEPLOYMENT IN THE WILD.

In addition to real-world results on Scannet++ 4.3, we also experiment with the deployment of `TST`, in a custom space. We collect a 15-minute video of a meeting room and used the resulting frames for pre-training described in Sec. 4.1 followed by a transfer on the ScanNet++ (Yeshwanth et al., 2023) transfer set (Sec. D). We evaluated `TST-MM` and the baselines on the semantic segmentation task. We evaluate `TST-MM` and the baselines on the semantic segmentation task. Tab. 3 shows that for this custom scene deployment, pre-training on the test-space through `TST-MM` outperforms the Internet-based baseline 4M-21 (Bachmann et al., 2024). The qualitative comparison in Fig. 15 shows that `TST-MM`'s predictions are notably better than those of the Internet-based 4M-21 (Bachmann et al., 2024).

| Method | mIoU |
|--------|------|
| Scratch | 21.82 |
| 4M-21 | 54.58 |
| `TST-MM` | **59.11** |

Table 3: **TST deployment in the wild.** Semantic segmentation performance comparison across training methods.

## G `TST` WITH THE DINOv2 OBJECTIVE.

In this section, we explore how `TST` trained with DINOv2 objective, `TST-DINO` from *scratch*, compares with its Internet counterpart trained on 142M images from the Internet (Oquab et al., 2023). Fig. 16 shows that pre-training on only data from the test space can substitute large-scale Internet pre-training. This further underscores that `TST` framework extends to other self-supervised

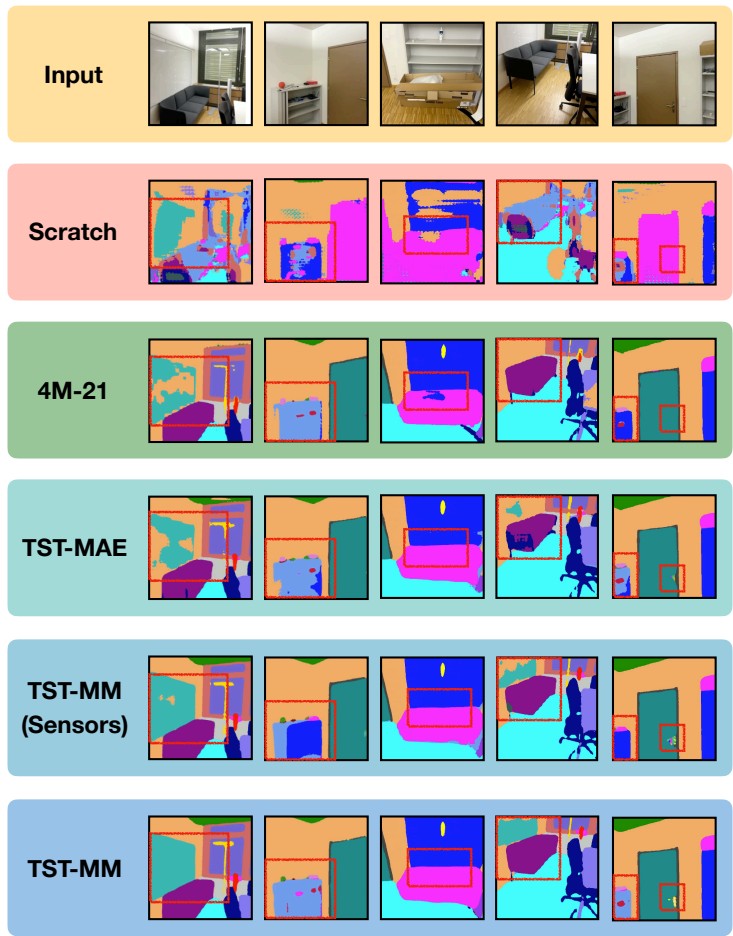

Figure 15: **TST-MM predictions on deployment in the wild**. We showcase the qualitative results for TST-MM on the semantic segmentation task against the Internet-based pre-trained model 4M-21(Bachmann et al., 2024) and scratch (no-pretraining). TST-MM predictions are notably better across object categories, showing the value of access to test space and the deployment potential of TST-MM.

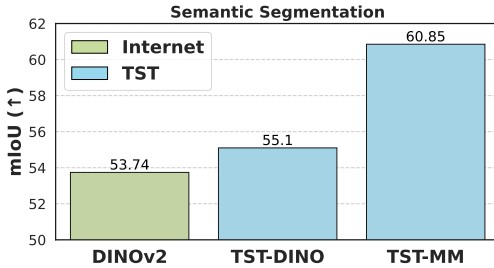
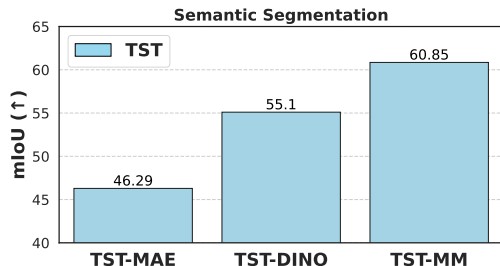

Figure 16: **TST with DINOv2 objective outperforms its Internet counterpart.** We compare the performance of DINOv2 pre-training in the test space, TST-DINO, with DINOv2 pre-trained on the large-scale Internet dataset of 142M images (Oquab et al., 2023). TST-DINO outperforms its Internet counterpart, showing the value of specialization. Yet, TST-MM with multimodal masked modeling achieves the best performance.

Figure 17: **Comparison between different pre-training objectives under the TST framework.** We compare the performance of different pre-training objectives using TST on the semantic segmentation task. We find that multimodal masked modeling (TST-MM) achieves the best performance followed by TST-DINO. All the three objectives were trained using the ViT-B model size on the ProcTHOR (Deitke et al., 2022) dataset.

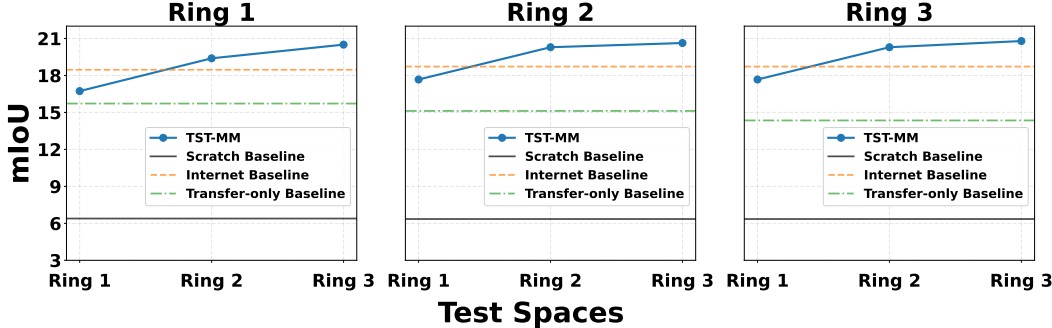

Figure 18: **Smallest unit of space to specialize on.** We reduce the test space size, that we can specialize and pre-train models with TST-MM. We compare it with an Internet pre-trained model (Bachmann et al., 2024), and a baseline that pre-trains only on the transfer set. We also find that training on a ring smaller than the test ring, leads to diminished performance.

objectives (Oquab et al., 2023) beyond masked modeling for specialization. However, we find that TST-MM, which uses multimodal masked modeling outperforms other unimodal self-supervised objectives like DINOv2 (Oquab et al., 2023) and MAE (He et al., 2021).

## H  BENCHMARKING DIFFERENT SELF-SUPERVISED OBJECTIVES UNDER TST.

We compare the performance of TST with different pre-training objectives such as multimodal masked modeling (Bachmann et al., 2024), DINOv2 (Oquab et al., 2023) and MAE (He et al., 2021). As shown in Fig. 17, we find that multimodal masked modeling (TST-MM) to be the most performant among the self-supervised objectives we explored. However, note that all 3 objectives show specialization trends as presented in Fig. 7 and Fig. 22.

## I  WHAT IS THE SMALLEST UNIT OF SPACE WE CAN SPECIALIZE ON?

In the results presented so far, we have shown that TST can specialize on test spaces at the size from 1-8 houses. However, this raises a question, what is the smallest unit of space we can specialize on? To probe this, we reduce the size of the test space and evaluate if TST can specialize to it. We consider a model trained via TST specialized, if it can outperform an off-the-shelf Internet-based generalist, when evaluated on that test space. We reduce the test space, in the form of concentric

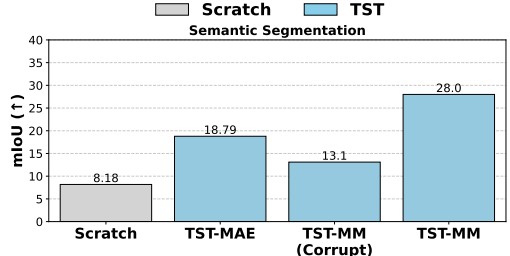
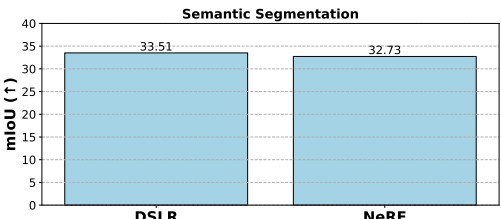

Figure 19: **TST with *untrained* pseudolabel modalities.** We train variants of `TST`, with RGB-only (`TST-MAE`), with RGB and 4 pseudolabel modalities (Depth, Surface Normal, CLIP and Imagebind), with trained networks (`TST-MM`), and with untrained or *corrupt* networks (`TST-MM (Corrupt)`). All results use a ViT-S backbone model on Scannet++.

Figure 20: **TST with synthetic data.** We replace real DSLR images in ScanNet++ (Yeshwanth et al., 2023) with NeRF (Mildenhall et al., 2020)-rendered images from the same training viewpoints. We find only negligible performance loss, demonstrating that NeRF output quality at known poses is sufficient to substitute high-quality DSLR images.

rectangles, starting with a room, and then reducing the size of the rectangle. For each rectangle, we pre-train a specialist model via `TST`. We compare this against 4M-21 (Bachmann et al., 2024), on the task of semantic segmentation. As shown in Fig. 18, we find that we can specialize on a single room (ring 3) that has an area of 20 square metres, and this trend continues as we reduce it down to ring 2, which is 11 square metres and ring 1 which is 5 square metres. Reducing the test space, below 5 square metres results in failed specialization, where the pre-training on just the transfer pre-training performs the best.

## J  DO WE NEED *trained* NETWORKS OUTPUTS FOR MODALITIES?

As described in Sec. 4.3, `TST-MM` leverages off-the-shelf pseudolabel networks to create additional modalities. This is akin to distilling their feature space on test space data, without directly accessing the external data that these networks were trained on. This raises an intriguing question, is it the trained feature space of a network Radford et al. (2021); Girdhar et al. (2023), that makes it a good modality, or simply the architecture inductive bias that creates a different transformation of the input image is sufficient to drive performance. To study this we present an analysis in Fig. 19, which shows that using untrained pseudolabels as modalities drops performance below the RGB-only baseline, suggesting that model capacity spent on learning those modalities does not lead to a useful representation.Thi suggests that *the learned feature space of pseudolabel network is crucial in making it as useful modality*.

## K  TST WITH SYNTHETIC DATA.

Recent advances in novel view synthesis (Mildenhall et al., 2020; Barron et al., 2021; Kerbl et al., 2023) have enabled realistic renderings of indoor spaces, opening up the potential for generating synthetic training data. In `TST`, we leverage existing indoor scene datasets (Yeshwanth et al., 2023; Straub et al., 2019), which include real RGB images captured with DSLR/iPhone cameras or rendered from 3D meshes, to develop specialized models for specific test spaces. This leads to a key question: if a novel view synthesis model can generate images from arbitrary viewpoints in a test space, can it serve as a controllable data generator—and can its outputs match real images in utility?

To explore this, we train a NeRF model (Tancik et al., 2023) using DSLR images from Scan-Net++(Yeshwanth et al., 2023), and render images from the same camera poses. We then pre-train two models—one using real DSLR images and the other using NeRF-rendered views—to assess the performance gap. As shown in Fig.20, NeRF-generated images result in negligible performance loss compared to real images. This suggests an interesting future direction: if high-fidelity NeRF models

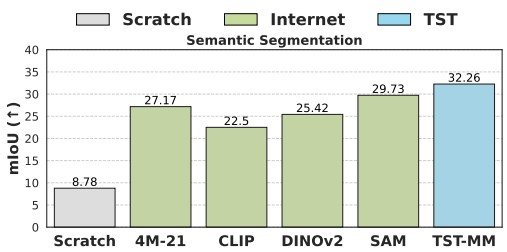

| | Test Space | Transfer | Segmentation (mIoU ↑) |
|---|---|---|---|
| | ✗ | ✓ | 42.01 |
| TST | ✓ | ✗ | 50.21 |
| | ✓ | ✓ | **56.96** |
| 4M-21 | | | 46.12 |

Table 4: **Ablating the use of transfer RGB frames during pre-training**. As noted in Sec. 4.1, we additionally use RGB images from the transfer set during pre-training. We ablate this choice by comparing all three dataset configurations. We use the ViT-S backbone for all models.

Figure 21: **`TST` works with off-the-shelf transfer set.** For Replica (Straub et al., 2019), we find that even when we use ADE20k (et al., 2017) as a transfer set, `TST-MM` outperforms Internet-based generalist models, showcasing the importance of having access to the test space, agnostic to the transfer set.

can be trained with fewer input images, they could act as steerable data generators, reducing the need for extensive real-world data collection in test environments.

## L  THE ROLE OF THE TRANSFER DATASET MIX-IN DURING PRE-TRAINING.

We study the role of mixing images from the test space and transfer datasets during pre-training, as mentioned in Sec. 4.1. Tab. 4 shows that using only test-space data outperforms both pre-training on large-scale Internet data and using only transfer images, but mixing test space and transfer data achieves the best performance. We hypothesize that seeing transfer images during pre-training helps the model to better align with the fine-tuning stage on the transfer dataset. Note that it cannot be explained by more diverse data in the transfer set, as adding non-test spaces decreases the specialization performance, as observed in Fig. 9.

## M  TST WITH OFF-THE-SHELF TRANSFER SET.

As noted in Sec. 4.1, for each dataset (Deitke et al., 2022; Straub et al., 2019; Yeshwanth et al., 2023) we explore, the transfer set comes from a similar domain, as the pre-training dataset, albeit from non-test spaces. It naturally raises the question, what if we use an existing off-the-shelf semantic segmentation dataset like ADE20k (et al., 2017) as a transfer set. Does `TST` generalize and result in performant specialist models, or is an in-domain transfer set necessary? To probe this, for the Replica (Straub et al., 2019) dataset, we pre-train `TST-MM`, but instead of using non-test spaces from Replica as the transfer set, we use ADE20k (et al., 2017). Fig. 21 shows `TST-MM` outperforms various generalist models (Bachmann et al., 2024; Oquab et al., 2023; Radford et al., 2021), even when using ADE20k (et al., 2017) as the transfer set. All models are evaluated in the test space from Replica (Straub et al., 2019), on semantic segmentation, with a ViT-B backbone.

## N  PSEUDO-LABELER BASELINES

As mentioned in Sec. 4.3, we use various off-the-shelf networks to pseudolabel RGB data, and create additional (optional) modalities for `TST-MM`. We present a comparison for `TST-MM` against these pseudolabel baselines in Tab. 5. `TST-MM` and `TST-MM` (adapted) outperform all pseudolabel baselines, suggesting the benefit of pre-training in the test space with them, via multimodal masked modeling.

| | Method | Semantic Segmentation | | | Object Detection | | Captioning | |
|---|---|---|---|---|---|---|---|---|
| | | Scannet++ mIoU ↑ | ProcTHOR mIoU ↑ | Replica mIoU ↑ | Scannet++ mAP ↑ | ProcTHOR mAP ↑ | ProcTHOR CIDEr ↑ | SPICE ↑ |
| Pseudo-labelers | ImageBind (Girdhar et al., 2023) | 25.40 | 44.54 | 12.78 | 6.78 | 32.54 | - | - |
| | CLIP (Radford et al., 2021) | 23.02 | 48.66 | 20.92 | 19.75 | 38.47 | 18.4 | 16.2 |
| | Mask2Former (Cheng et al.) | 29.42 | 50.28 | 22.68 | - | - | - | - |
| | ViTDet (Li et al., 2022) | - | - | - | 23.49 | 44.10 | - | - |
| | SAM (Kirillov et al., 2023) | 34.75 | 56.72 | 28.51 | - | - | - | - |
| Specialist Pre-training | TST-MM | 34.49 | **60.85** | 32.87 | 31.54 | 49.38 | 34.3 | 20.4 |
| | TST-MM (adapted) | **36.44** | 60.59 | **34.53** | **35.83** | **51.25** | **39.9** | **20.5** |

Table 5: **Comparing TST-MM against pseudolabels.** We find that TST-MM outperforms all pseudolabels, underscoring the value of pre-training on them via multimodal masked modelling in the test space.

## O  TST WITH NO SEMANTIC MODALITIES

TST-MM includes modalities obtained as outputs from different off-the-shelf models. Tab. 1 shows that TST-MM outperforms each individual model used as a modality. Since our transfer tasks are semantic segmentation and object detection, we further study if having off-the-shelf models trained on related tasks as modalities is crucial for our final performance.

We present three experiments using the ViT-B backbone on ProcTHOR (Deitke et al., 2022). For each experiment, we drop one of the following modalities: i) Semantic segmentation, ii) Object detection, iii) Semantic segmentation, Object detection, and SAM edges. Tab. 6 shows the results for each model when transferred to semantic segmentation and object detection. We find that even though the performance drops if we remove all three modalities, TST-MM still outperforms the Internet-based 4M-21 (Bachmann et al., 2024) model.

| Method | Modalities | | | | Task | |
|---|---|---|---|---|---|---|
| | Semantic segmentation | Object detection | SAM edge | Others | Segmentation (mIoU↑) | Detection (mAP↑) |
| TST-MM | ✓ | ✓ | ✓ | ✓ | **60.85** | 49.38 |
| | ✗ | ✓ | ✓ | ✓ | 59.43 | **49.58** |
| | ✓ | ✗ | ✓ | ✓ | 59.38 | 49.34 |
| | ✗ | ✗ | ✗ | ✓ | 55.39 | 45.97 |
| 4M-21 (Bachmann et al., 2024) | ✓ | ✓ | ✓ | ✓ | 53.24 | 41.43 |

Table 6: **The effect of semantic modalities in TST-MM.** As the results demonstrate, removing the semantic segmentation and object detection modalities obtained from off-the-shelf networks does not significantly hurt the TST-MM's performance on the downstream semantic segmentation and object detection tasks. When all three semantic modalities are removed, we observe a drop in performance, but TST-MM still outperforms the Internet-based 4M-21 (Bachmann et al., 2024) model, demonstrating the value of specialization.

## P  DO OTHER SELF-SUPERVISED OBJECTIVES BENEFIT FROM SPECIALIZED PRE-TRAINING?

In Sec. 4.3, we present results with TST-MM, which employs multimodal masked modeling. However, as mentioned in Sec. 3.2.2, TST also supports other self-supervised objectives. Fig. 22 shows that pre-training objectives, DINOv2 (Oquab et al., 2023), and RGB-only MAE (He et al., 2021) exhibit similar specialization trends.

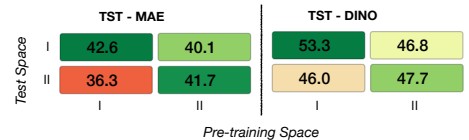

Figure 22: **Specialization using other objectives.** We demonstrate specialization using other pre-training objectives, including MAE and DINOv2.

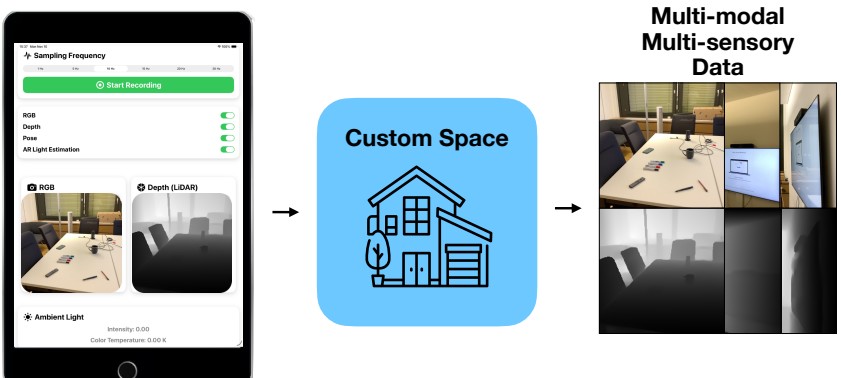

Figure 24: **iOS application for custom data collection.** It leverages the open source ARKit API to stream outputs of various sensors such as RGB, LiDAR, IMU, magnetometer, and ambient lighting, and supports paired data collection.

## Q  IS DISTILLING
## IN THE TEST SPACE BENEFICIAL?

As discussed in Sec. 4.3, we scale modalities by pseudo-labelling RGB data with various Internet-based models (Oquab et al., 2023; Radford et al., 2021; Cheng et al.). This process of creating additional modalities, and pre-training on them has enabled powerful multimodal foundation models (Mizrahi et al., 2023; Bachmann et al., 2024; 2022). This form of pre-training can also be seen as distilling the knowledge from these powerful off-the-shelf networks into a single unified model.

With `TST-MM`, we also distill from various off-the-shelf networks like CLIP (Radford et al., 2021), DINOv2 (Oquab et al., 2023) with masked modelling (He et al., 2021; Mizrahi et al., 2023). Results in Tab. 1, suggest that distilling with multiple modalities on the test space, results in performant specialist models. However, to disentangle the effect of multimodality and distillation, we take it one step further to probe whether just distilling in the test space, provides some additional benefit, over non-test spaces? Therefore, we distill, CLIP (Radford et al., 2021), DINOv2 (Oquab et al., 2023) and Mask2former (Cheng et al.) in the test space, and compare it with distilling in an IID, but non test space, and report the results in Fig. 23 on semantic segmentation in ProcTHOR (Deitke et al., 2022).

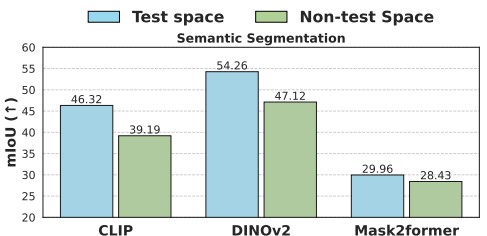

Figure 23: **Distillation in test space.** We find distilling over data from the test space, from various off-the-shelf models, results in more performant models in the test space. All results here are with the ViT-B backbone, on ProcTHOR (Deitke et al., 2022).

We find that distilling over data from the test space is more performant than the data from non-test spaces, underscoring the importance of access to the test space for specialization.

## R  APPLICATION FOR HARDWARE DATA COLLECTION

As discussed in Sec. 3.2.1, `TST` can be extended to leverage more hardware-based modalities, such as IMU, GPS, Audio, which can be found on most common user devices, such as iPhone. To facilitate future research in this area, we release an iOS application that enables anyone to collect aligned multimodal data from RGB-D and additional hardware sensors present on an iPhone. Fig. 24, shows an overview of our application.

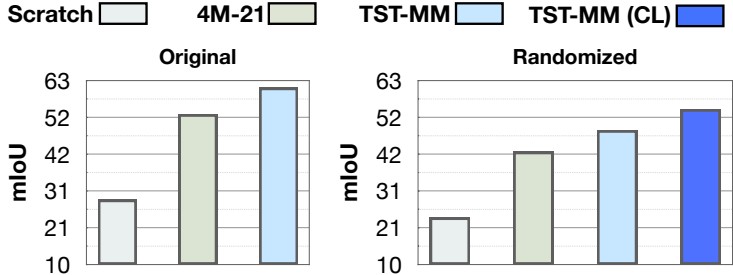

Figure 25: **Continual Learning with TST.** We study the performance of `TST-MM`, as the test space, undergoes lighting and minor object placement changes. The plot on the left, shows the result of the baselines on the original test space, without any changes. On the right, we present results after the test space has undergone lighting and object displacements. As expected, the `TST-MM` trained in the original test space, looses some performance, however as we continually train by collecting pre-training in the perturbed test space, we find that `TST-MM` (CL) quickly recovers performance.

## S  CONTINUAL LEARNING WITH TST.

As discussed before, when pre-training is performed on the exact same test space we deploy on, `TST` results in the most performant models. However, `TST` specializes in the test space, and all its characteristics, at the state when the pre-training data was collected. Therefore, a natural question to ask is, what happens if the test space undergoes some changes after data collection? This could include changes in the lighting of the space or minor object placements. We begin by investigating if these changes lead to a drop in performance for the `TST` model trained on the original test space. Thereafter, we leverage the ability of ProcTHOR to randomize object placements and lighting to create a perturbed version of the test space. Note that the overall layout and assets remain exactly the same, only the lighting and placement of small objects are varied.

We first evaluate the performance of `TST-MM` pre-trained on the unperturbed test space, on the perturbed test space (Fig. 25, right), and we find that it experiences a drop as compared to its performance in the original test space, (Fig. 25, left). However, as we continually pre-train the model by collecting data in the updated test space (`TST-MM` (CL)), it quickly recovers the loss in performance, and is still highly performant as compared to Internet-based generalists (Bachmann et al., 2024). This suggests that even under the condition that the test space undergoes changes, by simply continuing data collection in the test space, `TST` can continually improve its performance, without any access to external data.

## T  TEST TIME TRAINING WITH TST

As noted in Sec. 2, we share a similar goal with Test-time Training (TTT) (Sun et al., 2020) in bridging the train-test divide. TTT does it from the lens of inference time optimization to specialize to a particular test instance, whereas `TST` attempts to specialize to a given test space by pre-training in it.

However, in practice, these strategies can be orthogonal and complement each other. We can potentially apply TTT to a model pre-trained with `TST`, to improve its performance. To benchmark how this combination works, we conduct an analysis where we apply Test-time training with masked autoencoders (TTT-MAE) (Gandelsman et al., 2022a), with two pre-trained methods, MAE (He et al., 2021) pre-trained on Internet data and `TST-MAE` pre-trained on the test space.

| MAE (He et al., 2021) pre-training | Before TTT (mIoU ↑) | After TTT (mIoU ↑) |
|---|---|---|
| Internet | 34.54 | 39.28 |
| TST | **35.48** | **42.41** |

Table 7: **TST with Test-Time Training.** Before TTT corresponds to the performance of the models directly after the transfer training without any test-time training, whereas after TTT shows the results when test-time training on the test samples is performed. Both models use a ViT-B backbone and are evaluated on semantic segmentation on ProcTHOR (Deitke et al., 2022).

In TTT-MAE, we first start with a pre-trained MAE ViT-B encoder as the backbone, and train a task-specific head on the transfer set. During the test phase, the backbone is further tuned using the masked modeling objective for each test sample individually. This adaptive tuning enhances the model's performance on the downstream task for the given test samples.

As presented in Tab. 7, TTT-MAE improves the mIoU results for both the Internet pre-trained backbone and `TST-MAE`. However, we find that `TST-MAE` gets significantly more improvement than Internet-based MAE (He et al., 2021). Both models use a ViT-B backbone and are tested on semantic segmentation on the ProcTHOR (Deitke et al., 2022) dataset.

## U SAMPLING RATIO BETWEEN TEST SPACE AND TRANSFER DATA DURING `TST` PRE-TRAINING

As mentioned in Section 4.1, we found mixing RGB images from the transfer set into our pre-training data beneficial for performance. To study the interplay of this dataset mix further, we analyze the effect of the sampling frequency of the samples from the transfer set and the test space data during pre-training. A ratio of 1/1 implies that half the samples in pre-training come from the test space data and the other half from the transfer dataset. We pre-train the `TST-MM` model using both small and base sizes on the same test space as in Tab. 1, in the ProcTHOR (Deitke et al., 2022) dataset under different ratios. The models are then transferred and evaluated on the semantic segmentation task. Tab. 8

| Model Size | Transfer set / Test space set sampling ratio | | | | 4M-21 |
|---|---|---|---|---|---|
| | 1/1 | 1/4 | 1/8 | 1/16 | |
| Small | **61.01** | 59.03 | 56.96 | 57.01 | 46.12 |
| Base | 60.36 | 60.65 | **60.85** | 60.36 | 53.24 |

Table 8: **The effect of the sampling ratio between the test space and transfer data during pre-training.** We report transfer performance on semantic segmentation as we vary the sampling ratio between transfer and test space data during pre-training. There is no significant difference across ratios for the base model, while for the small model the best result is obtained with a one-to-one sampling ratio. Across all ratios, `TST-MM` outperforms Internet-based 4M-21 pre-training (Bachmann et al., 2024).

demonstrates the results for various ratio configurations and their effect on different model sizes. First, we find that in all cases, `TST-MM` consistently outperforms Internet-based 4M-21 (Bachmann et al., 2024) models of the same size. Secondly, we note that the performance of the bigger ViT-B based models is not sensitive to the ratio of sampling transfer and test space data, whereas for smaller ViT-S based models, a ratio of 1/1 seems to be a reasonable default choice.

## V PRIVACY ASPECT OF USING TEST-SPACE DATA.

`TST` presents a risk and an opportunity, in terms of data privacy, and we believe the opportunity could be more consequential. The risk stems from the data collected in the test space, which may need to be sent out for pre-training, as on-device compute may not be sufficient. While true, this does not seem insurmountable, as `TST` can ultimately enable training a compact, dedicated model for the user space, which may become possible with some in-house compute for sensitive domains.

On the other hand, continued commitment to the dominant paradigm of generalist models, trained on massive data, requires harvesting and pooling the data across a large number of users to create training datasets. This poses significant privacy risk — just not for the specific user that deploys the model, but for all the other users whose data had to be harvested to enable the creation of a generalist.

Taking further steps toward developing an effective test-space training paradigm with reduced need for external data, can decrease the necessity to pool diverse users' data to make large training datasets, thereby reducing the potential privacy abuse as well as unwanted commodification of data by corporate entities.

Besides the ethical aspect, it is interesting to more concretely understand the role of data in developing performant models (how much data, what kind of data, for pre-training or distillation, are all data equal, etc). The common assumption that large, diverse data in raw form (images and annotations) is

the way to go does not have to be the full story, as several papers, including Attention Transfer (Li et al., 2024a) and `TST` allude to.

# W  EXPERIMENTAL SETUP DETAILS

## W.1  PRE-TRAINING DETAILS

**Initialization.** For `TST-MM`, we use two initializations for pre-training. Unless stated otherwise, we pre-train our model from scratch, following the hyperparameters in Tab. 9. Additionally, for adaptation results in Tab. 1, we start from a pre-trained 4M-21 (Bachmann et al., 2024) model and finetune it with the hyperparameters in Tab. 10.

**DINO Pre-training.** For the DINO `TST` pre-training in Sec. P, we use the implementation from the original DINOv2 repository[7]. We use the default provided training configuration files and train a model with the ViT-B/14 backbone for 300,000 steps with a batch size of 1024.

| Configuration | Small | Base |
|---|---|---|
| Training length ($n$ tokens) | 100B | 500B |
| Warmup length ($n$ tokens) | 10B | |
| Optimizer | AdamW (Loshchilov & Hutter, 2019) | |
| Opt. momentum | $\beta_1, \beta_2 = 0.9, 0.95$ | |
| Base learning rate (Goyal et al., 2017) | 1e-4 | |
| Batch size | 4096 | |
| Weight decay | 0.05 | |
| Learning rate schedule | Cosine decay | |
| Feedforward activation | SwiGLU (Shazeer, 2020) | |
| Input token budget | 128 | 256 |
| Target token budget | 128 | 256 |
| Input and target $\alpha$ | Mixture (Bachmann et al., 2024) | |
| Masking strategy | Mixture (Bachmann et al., 2024) | |
| Image resolution | $224^2$ | |
| Augmentation | Random Crop | |
| Repeated sampling (Feichtenhofer et al., 2022) | 4 | |
| Data type | `bfloat16` (Burgess et al., 2019) | |

Table 9: **Pre-training settings for scratch initialization.** Training configuration for `TST-MM` initialized from scratch.

## W.2  TRANSFER DETAILS

**Semantic segmentation:** For semantic segmentation on ProcTHOR (Deitke et al., 2022), Replica (Straub et al., 2019) and Scannet++ (Yeshwanth et al., 2023) datasets, we use the ViT encoder from the pre-trained models with a decoder head, based on the ConvNext (Liu et al., 2022) network with a depth of 4. This decoder head is initialized from scratch. Training details are provided in Tab. 11. On Replica (Straub et al., 2019), and ProcTHOR (Deitke et al., 2022), pre-trained models are transferred and evaluated using a transfer training dataset of 20,000 images and evaluated on 5000 images sampled from the test space. On Scannet++ (Yeshwanth et al., 2023), we use a transfer dataset of 40,000 images and evaluated on 3000 images from the test space.

**Object detection.** For object detection, we evaluate pre-trained models by using the ViT-based pre-trained encoder as the feature extractor in the detection framework. We use Cascade Mask-RCNN (He et al., 2017; Cai & Vasconcelos, 2017) as our primary object detection model. Besides the feature extractor, the other learnable components including the detector's neck and head are initialized from scratch. All training and evaluations are performed using the Detectron2 (Wu et al., 2019) framework. Exact training settings are provided in Tab. 12. We evaluate object detection in the

---

[7]`https://github.com/facebookresearch/dinov2`

| Configuration | Small | Base |
|---|---|---|
| Training length ($n$ tokens) | 100B | |
| Warmup length ($n$ tokens) | 10B | |
| Optimizer | AdamW (Loshchilov & Hutter, 2019) | |
| Opt. momentum | $\beta_1, \beta_2 = 0.9, 0.95$ | |
| Base learning rate (Goyal et al., 2017) | 5e-5 | |
| Batch size | 4096 | |
| Weight decay | 0.05 | |
| Learning rate schedule | Cosine decay | |
| Feedforward activation | SwiGLU (Shazeer, 2020) | |
| Input token budget | 128 | 256 |
| Target token budget | 128 | 256 |
| Input and target $\alpha$ | Mixture (Bachmann et al., 2024) | |
| Masking strategy | Mixture (Bachmann et al., 2024) | |
| Image resolution | $224^2$ | |
| Augmentation | Random Crop | |
| Repeated sampling (Feichtenhofer et al., 2022) | 4 | |
| Data type | `bfloat16` (Burgess et al., 2019) | |

Table 10: **Pre-training settings for Internet initialization.** Pre-training configuration for TST starting from the the pre-trained 4M (Bachmann et al., 2024) model weights.

| Configuration | Small | Base |
|---|---|---|
| Fine-tuning epochs | 64 | |
| Warmup epochs | 1 | |
| Optimizer | AdamW (Loshchilov & Hutter, 2019) | |
| Opt. momentum | $\beta_1, \beta_2 = 0.9, 0.999$ | |
| Learning rate | 1e-4 | 2e-4 |
| Batch size | 32 (16 for Scannet++) | |
| Weight decay | 0.05 | |
| Learning rate schedule | Cosine decay | |
| Layer-wise lr decay (Clark et al., 2020) | 0.75 | |
| Drop path (Huang et al., 2016) | 0.1 | |
| Input resolution | $224^2$ | |
| Augmentation | RandomFlip + RandomCrop | |

Table 11: **Semantic segmentation settings.** Configuration used for fine-tuning the pre-trained models on the semantic segmentation task.

test spaces from the ProcTHOR (Deitke et al., 2022) dataset as described in Section 4.1. For transfer, we use a dataset of 20,000 images from an external space, that is different from the test space. We evaluate the transferred model on 5000 images from the test space.

| Configuration | Small | Base |
|---|---|---|
| Fine-tuning epochs | | 150 |
| Optimizer | | AdamW (Loshchilov & Hutter, 2019) |
| Opt. momentum | | $\beta_1, \beta_2 = 0.9, 0.999$ |
| Weight decay | | 0.1 |
| Learning rate | | 0.0001 |
| Learning rate schedule | | Multi-step decay |
| Lr schedule milestones | | [Epoch 133, Epoch 144] |
| Lr schedule decay values | | [1.0, 0.1, 0.01] |
| Warmup epochs | | 0.01 |
| Batch size | | 128 |
| Layer-wise lr decay (Clark et al., 2020) | | 0.7 |
| Drop path (Huang et al., 2016) | | 0.1 |
| Input resolution | | $224^2$ |
| Augmentation | | RandomFlip + RandomCrop |

Table 12: **Object detection settings.** Configuration used for fine-tuning the pre-trained models on the object detection task.

**Image Captioning.** For image captioning, we evaluate the pre-trained models obtained from various methods including `TST`, 4M-21 (Bachmann et al., 2024) (Internet), and also include randomly-initialized baselines (training from scratch). We adopt a standard transformer-based encoder-decoder architecture for image captioning and employ cross-entropy loss for next-token prediction during training. Images are input to the encoder which serves as the context for the decoder network. The encoder network is initialized from the respective method's encoder while the decoder is initialized randomly. Training and hyperparameter details are listed in Tab. 13. We also train a LLaVA style (Liu et al., 2023) model that serves as a Large-language-model-based baseline. We first train the connector module (MLP layer) using LLaVA's first-stage pretraining data consisting of 558K image-text pairs subset of the LAION-CC-SBU dataset (et al., 2021). For second-stage, we re-format our ProcTHOR captioning dataset into instruction-tuning format and jointly finetune both the connector and LLM.

**Captioning data generation.** To train models on the captioning task, we create a transfer dataset on a set of external spaces by generating captions using GPT-4o (OpenAI, 2023) for the transfer dataset. We follow a similar procedure for the evaluation set from the test space. We ensure the quality of generated captions by providing GPT-4o with multimodal inputs that include i) original RGB image ii) RGB image with instance-wise detection boxes and class names overlayed iii) Class names and bounding box coordinates in text format. We design an input prompt that instructs GPT-4o to leverage the multimodal inputs and generate COCO-style (Lin et al., 2014) 5 concise captions with global context per image. For a sanity check, we randomly sampled 500 generated samples from the transfer set and found all captions to be consistent with the visual contents present in their respective images. The prompt message used for generating captions from GPT-4o is shown in Fig. 26.

## X  COMPUTATIONAL RESOURCES.

All model pre-training and adaptations were done on 64 H100 GPUs, with the base and small models taking approximately 12 hours and 7 hours to train, respectively. For the semantic segmentation transfer runs, we fine-tuned the models on 4 H100 GPUs, resulting in approximately 3 hours of training for the base model and 1.5 hours for the small model. For the detection task, we only fine-tuned the base model on 8 A100 GPUs, training for approximately 6 hours. Similar to detection, for captioning we only fine-tuned the base model training on 8 H100 GPUs for approximately 6 hours.

```
I have a dataset of images captured in indoor settings showcasing different common household objects. I want to create
COCO-style concise and global captions for these images. Please generate a single caption for each image, adhering to
the following guidelines:

1. **Global Context but Concise**:
   The caption should be objective, describing the prominent objects and their spatial relationships within the scene.
Each caption must cover the global scene context and prominent objects.

2. **Use of Ground-Truth Classes**:
   Along with each image, ground-truth classes and bounding box information are provided. Bounding box information is
in the format `(upper left x coordinate, upper left y coordinate, width, height)`. Use bounding box information for
correct spatial relationships (such as left side, right side, top, below, etc.) between objects.

3. **Bounding Boxes and Class Labels Visualized in Image**:
   The bounding boxes and class names are overlaid on the image, showing each detected class for better localization.

4. **Spatial Positioning**:
   Describe all objects' positions and spatial relationships as visible in the image and ground-truth information to
help locate them accurately. If multiple objects are present in the image (as indicated in ground-truth information),
explicitly mention their count and explain their positional relationships with other objects in the image.

5. **No Hallucinations!**
   Each generated concise caption must agree with the actual contents shown in the provided image. Strictly avoid
adding information about objects unless you are certain. Only utilize the information visible in the image and the
provided ground-truth class information.

I will provide both the original image and the image with overlaid boxes and labels. Use both images to provide a
grounded global and COCO-style concise caption.

**Format your response**:
Return a Python list containing concise global captions. Do not output any other text.

Ground Truth information:  GT_class_and_bbox_information
Image with Annotations:  Image_Annotated
Original Image:  Image_Original
```

Figure 26: **LLM Prompt instruction for ProcTHOR caption generation transfer task**. We generate ground-truth captions by providing multimodal information to GPT-4o (OpenAI, 2023) including annotated image, class and instance-wise bounding-box information. For each image, we generate 5 COCO-style captions.

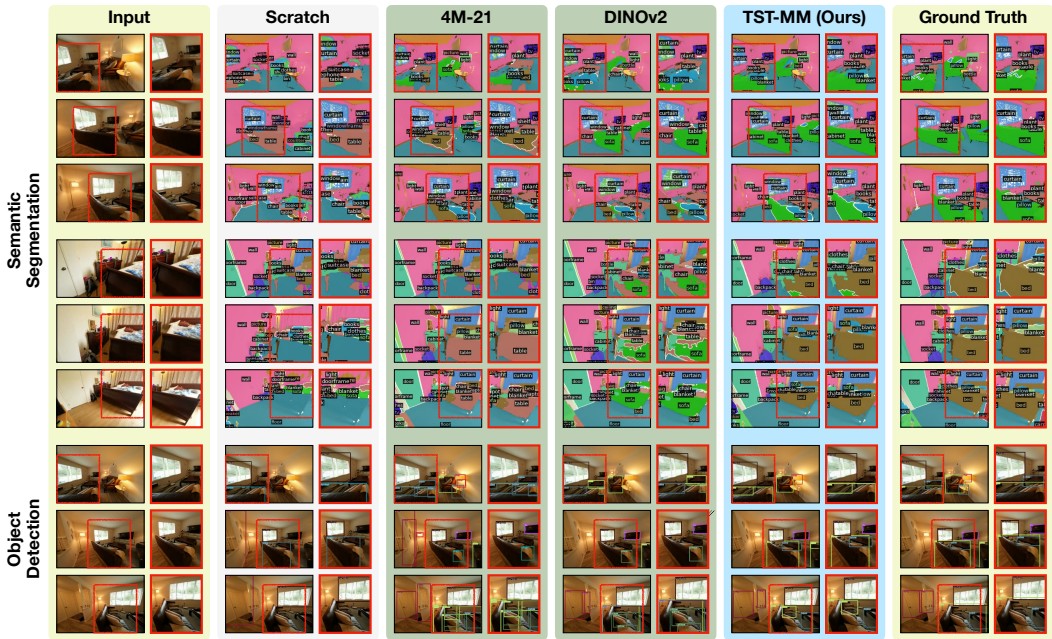

Figure 27: **Additional qualitative results.** As demonstrated here TST performs better compared to the other models for all tasks.

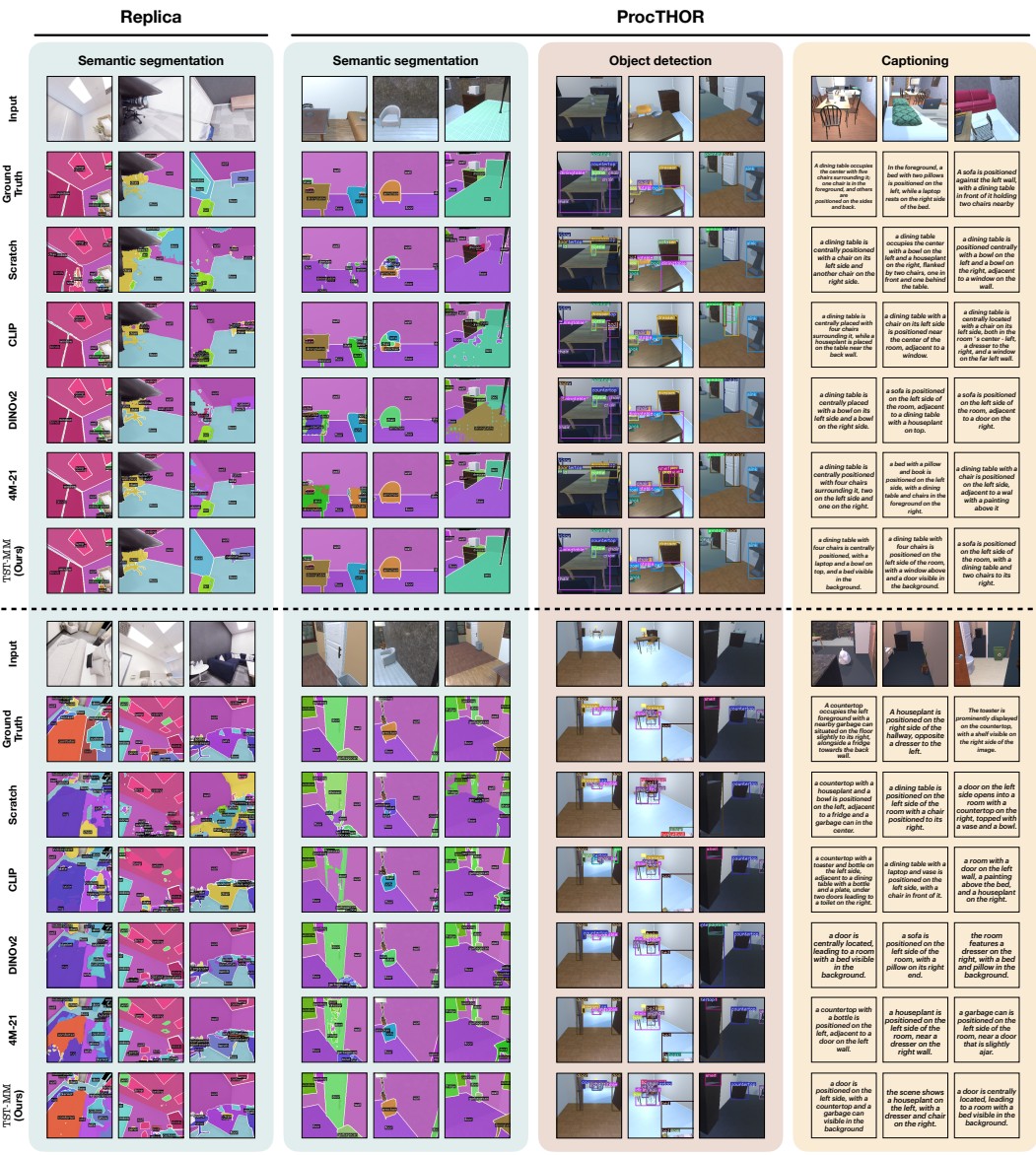

Figure 28: **Additional qualitative results.** As demonstrated here `TST` performs better compared to the other models for all tasks.

| Configuration | ProcTHOR Captioning |
|---|---|
| Fine-tuning epochs | 1400 |
| Warmup epochs | 600 |
| Optimizer | AdamW (Loshchilov & Hutter, 2019) |
| Opt. momentum | $\beta_1, \beta_2 = 0.9, 0.95$ |
| Base learning rate (Goyal et al., 2017) | 1e-5 |
| Batch size | 2048 |
| Weight decay | 0.05 |
| Learning rate schedule | Cosine decay |
| EMA decay | SwiGLU (Shazeer, 2020) |
| Eval. freq (epochs) | 50 |
| Input resolution | 224 |

Table 13: **Training details: Image Captioning.** Configuration used for transfer training for image captioning.

