# OpenReview forum: "MULTIMODALITY AS SUPERVISION: SELF-SUPERVISED SPECIALIZATION TO THE TEST ENVIRONMENT VIA MULTIMODALITY"
_ICLR.cc/2026/Conference — ICLR 2026 Poster_

### Official Review · Reviewer_r7Ks · 2025-10-29

**Soundness:** 3
**Presentation:** 3
**Contribution:** 3
**Rating:** 6
**Confidence:** 3

**Summary:**

This paper explores how rich multimodal data collected directly from the test environment can be used to pre-train visual representations in a self-supervised manner, without relying on any external data. To assess its effectiveness, the authors evaluate the proposed approach across multiple datasets and downstream tasks, showing that it can even outperform generalist models pre-trained on large-scale Internet datasets.

**Strengths:**

1. This paper proposes training a specialized model using rich multimodal data collected from the test space without relying on external data sources, which is highly practical and closely reflects real-world scenarios. Moreover, they provide an insightful analysis of the trade-off between specialization and generalization, as well as the effectiveness of specialization, demonstrating its practical value.

2. The viewpoint presented in lines 172–174 and fig. 4 is quite insightful. Nowadays, most research focuses on training large-scale generalist foundation models with massive datasets, but this paper highlights the often-overlooked importance of multimodality.

3. The paper is clearly written and easy to follow.

**Weaknesses:**

1. The authors pre-train their model on the entire dataset but evaluate it on only a small portion of it. For instance, in the Replica dataset, the model is pre-trained on 84,889 samples but evaluated on only 5,000 images. Could the authors clarify the rationale behind this evaluation setting and discuss whether it might affect the reported results?

2. In Table 2, the superscript “²” attached to Task-Specific Methods is referenced on page 8, but its explanatory note appears on page 7.

**Questions:**

Could the authors provide more explanation on why TST-MM performs less effectively on the captioning task?

---

> ### Author Response · Authors · 2025-11-22
> **Author response for r7Ks**
>
> Thank you for finding the raised problem formulation and study interesting. We respond to your questions below. We are revising the paper and disposition accordingly, which will be uploaded here in the coming days. We will be happy to discuss any follow-up comments.
>
> - **The authors pre-train their model on the entire dataset but evaluate it on only a small portion of it. For instance, in the Replica dataset, the model is pre-trained on 84,889 samples but evaluated on only 5,000 images. Could the authors clarify the rationale behind this evaluation setting and discuss whether it might affect the reported results?**
>
> 	- We found that, for a given test space, a test set of 3-5k images was large enough to yield significant results that do not vary when we increase the size of the inference set. The tables below show this on the Replica and Scannet++ datasets, where different test set sizes do not differ significantly.
>
> ### Replica Evaluation
>
> | # Eval Samples | mIoU  |
> |-----------------------------:|------:|
> | 5000                         | 32.87 |
> | 10000                        | 32.77 |
> | 15000                        | 32.80 |
>
>
>
> ### Scannet++ Evaluation
>
> | # Eval Samples |  mIoU |
> |--------------------------------:|------:|
> | 3000                            | 34.49 |
> | 10000                           | 33.37 |
> | 15000                           | 33.57 |
>
> - **Table 2, explanatory footnote appears on the next page**
>
> 	- Thanks for pointing this out. We will update the formatting to correct these in the final draft.
>
> - **TST-MM on image captioning**
>
> 	- During pre-training, TST-MM does not see any text data, as opposed to baselines like 4M-21 and CLIP, which are pre-trained on text data. Additionally, task specialist baselines like LLaVa [1] leverage an LLM to produce text, as opposed to TST which finetunes the same transformer decoder used for multimodal pre-training, with no text modality. Therefore, it is expected that the model performs less effectively on this task as there is a vast gap to fill. However, note that in spite of not seeing any text in pre-training, TST-MM, from scratch, performs on par with 4M-21, which was pre-trained on large-scale image-text data [2]. And additionally, when we adapt 4M-21 via TST, resulting in TST-MM (Adapted) (Fig 2), it becomes competitive with LLaVa [1], and outperforms other generalist models like CLIP.
>
> References
>
> [1] LLaVA: Large Language and Vision Assistant. Liu*, Li* et al, 2023
>
> [2] CC12M, Conceptual 12M: Pushing Web-Scale Image-Text Pre-Training To Recognize Long-Tail Visual Concepts. Changpinyo et al, 2021

---

> > ### Comment · Reviewer_r7Ks · 2025-11-22
> >
> > Thank you for your efforts. My comments have been adequately addressed and I decided to increase my score.

---

### Official Review · Reviewer_ygUd · 2025-10-31

**Soundness:** 3
**Presentation:** 1
**Contribution:** 4
**Rating:** 6
**Confidence:** 4

**Summary:**

The paper tackles an important problem, how to use data from a given deployment scenario effectively in the pre-train and post-train phase to maximally improve performance in that specific scenario we care about -- this would be the central challenge in deploying robotics applications in the real world.

The proposed method is straightforward, they use a self-supervised multi-modality objective to pre-train the model on the test space only. Additionally, they use features from web-scale models as additional self-supervisory alignment targets for improving performance, showing that direct access to web-scale data while training is unnecessary and methods like Attention Transfer are enough. The authors further show more analysis of the specialization-generalization tradeoffs, paving the way for personalization of models for each deployment site through their simple training framework.

**Strengths:**

In general, a well-executed paper with thoughtful experiments targeting a real problem.

1. The idea of employing test-space data to personalize the model to a given deployment scenario is feasible and very useful in many real world situations.
2. TST-Adaptation and TST-Sensory methods make sense to me, glad to know self-supervised cross modality helps as a self-supervised objective.
2. Very useful ablations targeting the real deployment problems that would plague perception and robotics in the coming years. Namely, "How many spaces is one test space worth?" and the "specialization-generalization tradeoffs" are valuable signals and will become increasingly critical for robotics deployment in the coming years.

**Weaknesses:**

1. Parts of the paper over claims w.r.t the TST-MM method.

(a) TST-MM is trained on additional pseudo modalities obtained by executing generalist models on the test space data. If we employ pseudo labelling from generalist models like CLIP, ImageBind and SAM, this is akin to distilling features (i.e. some spiritual variant of Attention Transfer [a]) of the generalist models, thus utilizing the pre-trained nature of the generalist model trained on internet data. This is framed as "psuedo-modality" instead of calling the spade what a spade is -- i.e. leveraging web-scale model's generalist pre-trained feature space.

(b) The central claim that "access to" generalist data is unnecessary, while true, is riddled with caveats. The specific writing felt misleading as it seemed to imply that web-scale data/models are not needed and the caveat should be clarified at the outset in the abstract, teaser figure and introduction. As the paper itself shows, It is definitely necessary to have access to the generalist models (and indirectly their data and learned manifolds) to obtain the pseudo modalities for training.

(c) The phrasing of the abstract and introduction seem to paint a picture that web scale datasets (and models) are not needed and generalist pre-training is not needed -- this is not true as it's indirectly used by this method. Please clarify what am I missing here, maybe the key claim should be rephrased to say "one does not need to 'directly' access and train on web-scale data while deploying models to a specialized test space, feature alignment from such generalist models is enough".

2. TST-Sensory: How does it compare with TST-MM (i.e. with the web scale targets as modality sources)?

3. When the generalist models (DINOv2, CLIP etc) are trained on the small external dataset, is the generalist backbone frozen? Many vision works have shown that -- surprisingly -- finetuning the DINOv2 backbone is counter-productive.

4. No discussion of privacy concerns when using test space data. Is it possible to do this form of (pre/post-)training on-premise/on-device efficiently. For example, How do we ensure that data from deployment is minimally accessed by a robotics developer due to heightened privacy concerns of accessing such test space data. In the considered scenario, house hold robots would collect data of end-users engaging in day to day activities, I'm not sure how all the stakeholders would react to their private data being accessed over the internet, and what ethical, technical and sociological frameworks are necessary to address issues arising from the concerns of such stakeholders. While I don't expect this specific paper to solve this problem completely, a discussion or feasibility study keeping these concerns in mind would significantly strengthen this paper given it's applicability to real world deployment.

[a] On the Surprising Effectiveness of Attention Transfer for Vision Transformers, NeurIPS 2024

**Questions:**

Please clarify the points made in weaknesses. I like this paper overall and would be open in increasing my rating further, however, the writing in a few places feels misleading to me and paints a picture that would border on over-claiming in my view (maybe the terminology needs to be refined in this case) -- please list down steps that would make the claims in the abstract and introduction much more tighter and easier to digest.

---

> ### Author Response · Authors · 2025-11-22
> **Author response to review ygUd, part 1**
>
> We are genuinely grateful for the thought-through, meaningful review. We enjoyed reading it. In short, we are fully aligned with all your major points. We expand on that below, and we are revising the paper and disposition accordingly, which will be uploaded to open review in the coming days. We will be happy to discuss any follow-up comments.
>
> - **"TST-MM leveraging internet-based pseudo-modalities: pseudo-labeling from generalist models like CLIP is akin to distilling features of the generalist models, thus utilizing the pre-trained nature of the generalist model trained on internet data. The key claim should be rephrased to say "one does not need to 'directly' access and train on web-scale data while deploying models to a specialized test space, feature alignment from such generalist models is enough"".**
>
> 	- This conclusion is entirely correct, and we are in agreement. This was the intention of experimenting with a “Sensory only” setup (Section 4.2) to have a reduced case that truly needs no external data, directly or indirectly, as it uses device sensors to get its modalities. We also recognized that pseudo-labeling is essentially the distillation of pre-trained generalist models and wrote Section R (and Figure 20) of the Appendix on that, but we understand that some of these points are buried deep in a long Appendix, though we had made the connection and experimented around it. We also agree that the current writing conflates these aspects and needs to be revised to convey the actual picture. Let’s establish a consolidated picture:
>
>
> 	- Here, we show the relevant methods on one spectrum. This is for an experiment on semantic segmentation with Scannet++[3], with a ViT-B backbone. The lower bound is scratch (i.e., no pre-training and learning using the external transfer set only). The upper bound is approximated by a fully supervised model with the same architecture as others and trained with a large number of annotated segmentation images. The gap between the lower and upper bounds is the playfield for the pre-training methods to fill.
>
>  ```
>                Multimodal   TST-MAE         TST-Sensors
>                Scratch
>                 7.82        17.62           28.67
> ~Lower bound ●-●------●----●--------------●●--------●---------------------------------------● ~Upper bound
>               7.49     13.74               27.59     34.49                                   62.03
>               Scratch  MAE                 4M-21     TST-MM                                  Fully Supervised
>                                                                                              Large Data
>
> Scannet++, ViT-B backbone, Semantic Segmentation
> ```

---

> ### Author Response · Authors · 2025-11-22
> **Author response to reviewer ygUd, part 2**
>
> Observations:
>
> 1. As you can see, TST-Sensors covers over 1/3 of the way; therefore, it does provide a nontrivial value.
> 2. However, training with sensory modalities is not a sufficient condition for the emergence of semantics — at least given the current sensory modalities, the current multimodal training objective, and the current inductive biases in the architecture. This is expected. Enriching the modalities with rich pseudo-labelers significantly improves results (TST-MM), leading to the best performing method and covering half of the way. This model no longer qualifies to claim no access to external data, as it uses pseudo-labelers trained on external data, akin to distilling them (albeit distilled on test-space data only).
> 3. Yet, an observation worth noting is that the distillation of such pseudo-labelers is more effective when done on the test-space data vs external data. This is what comparing TST-MM with 4M-21 shows, as the two are equivalent in nearly all aspects, except that 4M-21 distills the pseudo-labelers on external data (CC12M [2]) while TST-MM distills the same pseudo-labelers only on a much smaller test-space data. The same observation was consistently made for various settings in Figure 20 of the Appendix.
> 4. In general, multimodality is useful as the multimodal models consistently outperform their single-modal counterparts (e.g., TST-MM vs TST-MAE. Or 4M-21 vs MAE).
> 5. The same spectrum figure can be made using other datasets and settings (e.g. shown below for ProcTHOR, with a ViT-S backbone). They show the same picture: a) the sensory-only model does cover a substantial way, b) the richer version benefiting from generalist pseudo-labellers performs better and is the leading model, c) multimodality is useful, d) distillation on test-space data is more effective than external data.
>
> ```
>                                             4M-21             TST-MM
>                                             46.12             56.96
> ~Lower Bound ●-------------------------●----●---------●-------●-----------------------------● ~ Upper Bound
>              26.70                     42.68          52.27                                 75.14
>              Scratch                   TST-MAE        TST-Sensors                           Fully Supervised
>                                                                                             Large Data
>
>
> ProcTHOR, ViT-S backbone, Semantic Segmentation
> ```
>
> Also, we would like to note that we view the studied setup on specialization and test-space training more as a “sandbox” and problem formulation, rather than as a method that solves the problem fully now. As mentioned earlier, the current models are doing something useful, but the current multimodal training objectives and the architecture's inductive biases are clearly suboptimal. There is a lot of room to strengthen them (by improving the objective and modalities to include multi-viewness, motion, a notion of unsupervised objectiveness, multi-agent observations, etc.). It’s the goal of future research to advance the methodology so that the TST variants inch toward the upper bound and ultimately find a setup that ideally closes the gap without needing substantial external information. Along these lines, we showed in Fig. 21 of the Appendix that we have developed a software suite to record rich sensory modalities from common devices, namely iPhone/iPad Pros, as a support toward this goal, which we will open-source with the paper.
>
>
> - **Paper updates**: We will clarify and revise the paper in accordance with the above discussion. Specifically,
> 1) We’ll put the above consolidated results and discussion for a clear full picture in the paper early on
> 2) We’ll revise the claims to clarify that external data is indeed needed, in feature distillation form, to achieve the best results; though the variant that the sensory-only variance that does not use external data still makes meaningful gains
> 3) We’ll raise the discussion that the access to the external data does not have to be “direct”, but in feature distillation form, similar to other papers in the community such as attention transfer [1]. And that the distillation shows interesting trends, as distilling on test-space data is more effective than external data.

---

> ### Author Response · Authors · 2025-11-22
> **Author response to reviewer ygUd, part 3**
>
> - **"How does TST-sensors, compare to TST-MM?"**
>
> 	- The spectrum presented in the above response, presents a comparison between TST-Sensors and TST-MM.
>
> - **"Privacy concerns when using test-space data"**
>
> 	-   The privacy aspect of the studied formulation is interesting. It creates both a risk and an opportunity, and we think the opportunity could be more consequential. The risk is that, as you pointed out, the test-space data needs to be trained on, and therefore may need to be sent out. While true, this does not seem insurmountable, as a specialization setting can ultimately enable training a compact, dedicated model for the user space, which may become possible with some in-house compute for sensitive domains. Especially since it’s fair to expect that training a specialized model will be much more efficient than training large generalist models on internet-scale data. On the other hand, if we continue to commit to the dominant paradigm of generalists trained on massive data, we are essentially requiring the data of many users to be harvested and pooled to create such training datasets, which poses its own privacy risk — just not for the specific user that deploys the model, but for all the other users whose data had to be harvested to enable the creation of a genearlist. If the field takes further steps toward developing an effective test-space specialization paradigm with reduced need to external data, we will decrease the necessity of pooling different users’ data to make large training datasets, which reduces the potential of privacy abuses as well as unwanted commodification of people’s data by large companies. Besides the ethical aspect, it is interesting to more concretely understand the role of data in developing performant models (how much data, what kind of data, for pre-training or distillation, are all data equal, etc). The common assumption that large, diverse data in raw form (images and annotations) is the way to go does not have to be the full story, as several papers, including Attention Transfer [1] and this submission, allude to.
> 	-  We had included a discussion in the Ethics Statement (Section 6) that we will expand with the above discussion. Thanks for showing interest.
>
> - **Label Propagation**
>
> 	- The experiments in the paper all use external transfer images — i.e., the annotated images used for training the transfers are from other spaces, rather than the test space. This is a sensible choice to match a real-world setup, as the user would not want to spend time annotating images of their space. However, this introduces a minor suboptimality in quantifications, since the transfer set may exhibit distribution shifts from the test space (e.g., if some test objects do not appear in the transfer set or have vastly different appearances). Therefore, it will not be easy to attribute poor results primarily to ineffective pre-training. To further isolate this aspect for a more concrete analysis, we experimented with a controlled setup in which the transfer images are a small set (~20) from the same test space. This is akin to having a few annotated images from the test space and expecting the model to propagate the labels to the rest of the images of the test space (hence the name “label propagation”). In this setting, any remaining gap can be attributed to the suboptimality of pre-training methods, as the transfer set is distribution shift-free.
> 	- Below are the results of this controlled setup, which paint the same picture as the previous results and provide a reassuring consistency.
>
> ```
>                           MAE    TST-MAE
>                           31.05  37.98
> ~Lower Bound ●-----------●●------●-●-----●-----●--------------------------------------------● ~Upper Bound
>              19.39       30.57     39.81 45.80 51.25                                        91.81
>              Scratch   Multimodal  TST   4M-21 TST-MM                                       Fully Supervised
>                        Scratch     Sensors                                                  Large Data
>
> Scannet++, ViT-B, Semantic Segmentation (label propagation)
> ```
>
>
> References
>
> [1] On the Surprising Effectiveness of Attention Transfer for Vision Transformers. Li et al, 2024
>
> [2] CC12M, Conceptual 12M: Pushing Web-Scale Image-Text Pre-Training To Recognize Long-Tail Visual Concepts.
> Changpinyo et al, 2021
>
> [3] ScanNet++: A High-Fidelity Dataset of 3D Indoor Scenes. Yeshwanath*, Liu* et al, 2023

---

> ### Comment · Reviewer_ygUd · 2025-11-24
> **Looks good to me**
>
> Thanks for the response, I'll be raising my rating, hopefully this work opens new avenues and challenges for the community to tackle. Please add the "text" figures as actual figures in the paper, and fix the framing of the abstract and introduction. All the best!

---

> > ### Author Response · Authors · 2025-11-27
> > **Authors' Follow up to reviewer comment**
> >
> > Thank you for your response and support for our work. We will update the paper for the camera-ready version, with the discussion presented above.

---

### Official Review · Reviewer_NjHj · 2025-10-31

**Soundness:** 3
**Presentation:** 3
**Contribution:** 2
**Rating:** 4
**Confidence:** 3

**Summary:**

This paper proposes Test-Space Training (TST), a self-supervised framework that learns visual representations directly from multimodal data within the deployment environment, without using any external datasets. By leveraging cross-modal masked modeling among locally available sensors (e.g., RGB, depth), TST treats multimodality as supervision to pre-train models specialized for the target test space. Experiments across different datasets and various downstream tasks show that TST consistently matches or surpasses large-scale Internet-pretrained models (e.g., CLIP, DINOv2, 4M-21).

**Strengths:**

The paper explores an interesting and meaningful problem. It is clearly written and well-structured, making the methodology and insights easy to follow.

**Weaknesses:**

1. The paper raises an interesting problem but lacks methodological innovation — the proposed TST framework mainly integrates existing components, with less than one page describing the method in detail.
2. Inference relies on multimodal inputs, yet potential modality bias or inconsistency is not addressed; the paper should clarify how multimodal information is effectively fused for downstream tasks.
3. In Figure 3, TST-MM still performs noticeably worse than large-scale Internet-based pre-training, and the paper should analyze the reasons behind this gap.
4. There are several formatting issues: for instance, the description of Table 1 should appear on the same page, and figure order needs adjustment (e.g., Figures 6–8 appear out of sequence).
5. The meaning of different colors in Figure 8 is not explained and should be clarified.

**Questions:**

1. The proposed TST framework mainly combines existing components with limited methodological detail. Could the authors clarify what the core technical novelty of TST.
2. During inference, different modalities may introduce cross-modal bias or inconsistency. How does the proposed method effectively handle or align such discrepancies.
3. In Figure 3, TST-MM still performs worse than large-scale Internet-pretrained models. Could the authors provide an analysis or discussion explaining the potential causes of this performance gap.
4. In Figure 8, the meaning of different colors is not clearly explained. Could the authors specify what each color represents.

---

> ### Author Response · Authors · 2025-11-22
> **Author response to reviewer NjHj, part 1**
>
> We thank the reviewer for the comments and questions. We respond to your concerns below. We are revising the paper and disposition accordingly, which will be uploaded here in the coming days. We will be happy to discuss any follow-up comments.
>
>  - **Methodological innovation**
>
> 	 - The architecture we adopted is indeed an existing, widely used encoder-decoder one based on masked modeling. However, an architectural novelty was not the goal of this paper, and it provides several other aspects that qualify as interesting novelty, as pointed out by reviewers ygUd and r7Ks as well. For example, 1) raising the Test-Space Training problem definition, 2) demonstrating multimodality as an effective supervision for this purpose, 3) a diverse set of detailed studies to extract insights, e.g., modality scaling vs data scaling (Sec 4.3 & Fig 4, Fig 5), specialization-generalization trade-off (Sec 4.4 & Fig 7), test-space training using other self-supervised objectives (Fig 19, Fig 20). Two points are expanded as examples:
> 	    -   **Scaling modality vs Scaling data:** Recent advances in machine learning often rest on the common foundation of scaling data and its diversity with the target of creating a generalist model. Scaling datasets have enabled remarkable generalization abilities; for instance, CLIP was trained on 400 million image-text pairs, and DINOv2 is trained on 142 million examples from LVD142M. In this work, we explore an alternative paradigm, where we constrain the evaluation to just one test space and study the effect of _scaling the number of modalities_, as opposed to scaling unimodal data from external sources (Figure 5). This paradigm for scaling has not been concretely inspected before, and serves as a useful starting guide for future studies and for various downstream applications.
> 	    -   **Multimodality was key for building more performant models in the test space:** TST enables a highly performant model (Table 1, Figure 3) for a given test space by pre-training on data collected from the test space. However, is it _obvious_ that simply training on the data from a given test space will yield the most performant model for that space, even outperforming popular off-the-shelf models? We highlight that it is not the case, and _multimodality as supervision, not just using test-space data, was essential to achieve performant results_. Concretely, TST-MAE does not outperform internet-based pre-training methods. However, TST-MM, which pre-trains on the exact same data, but with a richer set of modalities (Section 4.5), outperforms all internet-based pre-training methods (See Table 2).
>
>
> - **In Figure 3, TST-MM still performs worse than large-scale Internet-pretrained models. Could the authors provide an analysis or discussion explaining the potential causes of this performance gap**
>
>
> 	- Note that in Figure 3, the model is “TST-MM (Sensors only)” — a reduced variant of the model that has access to only 4 modalities: RGB, Depth maps, Surface Normals, and Canny Edges. This experiment was intended to quantify whether a performance gain is possible even if the model doesn’t have access to pseudo-labelers for a richer set of modalities. As the figure indicated, the answer was yes. This model performs on par with DINOv2 on semantic segmentation, and slightly underperforms on object detection. The more performant version of our model, TST-MM (as presented in Table 2), has a richer set of modalities and, as expected, outperforms all baselines, including internet-based generalist and task-specialist models (described in Section 4.5). We will make sure this is clear in the writing.
> 	-   For a more detailed understanding of this trend, the modality scaling curve (Section 4.3, Figure 5) studies how the performance of TST scales as we increase the set of modalities.
>
>
> - **Inference relies on multimodal input, introducing potential modality bias**
>
> 	- We do not use any multimodal inputs during inference. We use multimodality during training as self-supervision. Once we have a pre-trained model, for transfer learning, we only use RGB images as input to fine-tune and evaluate the model. Therefore, potential discrepancies across modalities will not affect the model at transfer or inference time.
> 	- We will add a clarifying line in Section 4.1, Baseline, stating this clearly as follows:
> 	 > We finetune the baselines with the same transfer data D_t, and use only RGB images as input for transfer and evaluation (except for the multimodal scratch baseline). An extensive hyperparameter search was done to ensure fair comparison.
> 	>
> 	> -   Note that for multimodal scratch baseline, we use an ViT encoder that can accept multiple modalities as input akin to 4M. We do not perform any explicit modality fusion.

---

> > ### Author Response · Authors · 2025-11-22
> > **Author response to reviewer NjHj, part 2**
> >
> > -   **Formatting issue, description for Table 1**
> >     -   Thanks for the pointer. We will correct this in the final version.
> >
> > -   **Colors in Figure 8**
> >     -   The colors represent simply the relative performance. The greener boxes indicate higher performance, and the more red ones indicate lower performance. We will update the caption to reflect this.

---

> ### Author Response · Authors · 2025-11-27
> **Author's follow up on initial response, part 1**
>
> Dear reviewer NjHj, please let us know if we can answer any further questions about our work. The other two reviewers are strongly supportive of the findings in this paper (with scores of 10 and 8). We want to hear from you on how we can make our paper stronger.
>
> To recap the other discussion, we summarize a consolidated picture here.
>
> - We present the relevant methods on one spectrum below. This is for an experiment on semantic segmentation with Scannet++[3], with a ViT-B backbone. The lower bound is scratch (i.e., no pre-training and learning using the external transfer set only). The upper bound is approximated by a fully supervised model trained with a large number of annotated segmentation images. The gap between the lower and upper bounds is the playfield for the pre-training methods to fill.
>
>  ```
>                Multimodal   TST-MAE         TST-Sensors
>                Scratch
>                 7.82        17.62           28.67
> ~Lower bound ●-●------●----●--------------●●--------●---------------------------------------● ~Upper bound
>               7.49     13.74               27.59     34.49                                   62.03
>               Scratch  MAE                 4M-21     TST-MM                                  Fully Supervised
>                                                                                              Large Data
>
> Scannet++, ViT-B backbone, Semantic Segmentation
> ```
>
> Observations:
>
> 1. As we can see, TST-Sensors covers roughly 1/3 of the way; therefore, providing a nontrivial value.
> 2. However, training with sensory modalities is not a sufficient condition for the emergence of semantics — at least given the current sensory modalities, the current multimodal training objective, and the current inductive biases in the architecture. This is expected. Enriching the modalities with rich pseudo-labelers significantly improves results (TST-MM), leading to the best performing method and covering half of the way. This model no longer qualifies to claim no access to external data, as it uses pseudo-labelers trained on external data, akin to distilling them (albeit distilled on test-space data only).
> 3. However, an observation worth noting is that the distillation of such pseudo-labelers is more effective when done on the test-space data vs external data. This is what comparing TST-MM with 4M-21 shows, as the two are equivalent in nearly all aspects, except that 4M-21 distills the pseudo-labelers on external data (CC12M [2]) while TST-MM distills the same pseudo-labelers only on a much smaller test-space data. The same observation was consistently made for various settings in Figure 20 of the Appendix.
> 4. In general, multimodality is useful as the multimodal models consistently outperform their single-modal counterparts (e.g., TST-MM vs TST-MAE. or 4M-21 vs MAE).
> 5. The same spectrum figure can be made using other datasets and settings (e.g. shown below for ProcTHOR, with a ViT-S backbone). They show the same picture: a) the sensory-only model does cover a substantial way, b) the richer version benefiting from generalist pseudo-labellers performs better and is the leading model, c) multimodality is useful, d) distillation on test-space data is more effective than external data.
>
> ```
>                                             4M-21             TST-MM
>                                             46.12             56.96
> ~Lower Bound ●-------------------------●----●---------●-------●-----------------------------● ~ Upper Bound
>              26.70                     42.68          52.27                                 75.14
>              Scratch                   TST-MAE        TST-Sensors                           Fully Supervised
>                                                                                             Large Data
>
>
> ProcTHOR, ViT-S backbone, Semantic Segmentation
> ```

---

> > ### Author Response · Authors · 2025-11-27
> > **Author's follow up on initial response, part 2**
> >
> > - **Privacy aspect of Test-Space Training**
> >
> > 	-   The privacy aspect of the studied formulation is interesting. It creates both a risk and an opportunity, and we think the opportunity could be more consequential. The risk is that, as you pointed out, the test-space data needs to be trained on, and therefore may need to be sent out. While true, this does not seem insurmountable, as a specialization setting can ultimately enable training a compact, dedicated model for the user space, which may become possible with some in-house compute for sensitive domains. Especially since it’s fair to expect that training a specialized model will be much more efficient than training large generalist models on internet-scale data. On the other hand, if we continue to commit to the dominant paradigm of generalists trained on massive data, we are essentially requiring the data of many users to be harvested and pooled to create such training datasets, which poses its own privacy risk — just not for the specific user that deploys the model, but for all the other users whose data had to be harvested to enable the creation of a genearlist. If the field takes further steps toward developing an effective test-space specialization paradigm, we will reduce the necessity of pooling different users’ data to make large training datasets, which reduces the potential of privacy abuses as well as unwanted commodification of people’s data by large companies. Besides the ethical aspect, it is interesting to more concretely understand the role of data in developing performant models (how much data, what kind of data, for pre-training or distillation, are all data equal, etc). The common assumption that large, diverse data in raw form (images and annotations) is the way to go does not have to be the full story, as several papers, including Attention Transfer [1] and this submission, allude to.
> > 	-  We had included a discussion in the Ethics Statement (Section 6) that we will expand with the above discussion. Thanks for showing interest.
> >
> > - **Label Propagation**
> >
> > 	- The experiments in the paper all use external transfer images — i.e., the annotated images used for training the transfers are from other spaces, rather than the test space. This is a sensible choice to match a real-world setup, as the user would not want to spend time annotating images of their space. However, this introduces a minor caveat in quantifications, since the transfer set may exhibit distribution shifts from the test space (e.g., if some test objects do not appear in the transfer set or have vastly different appearances). Therefore, it will not be easy to attribute poor results primarily to ineffective pre-training. To further isolate this aspect for a more concrete analysis, we experimented with a controlled setup in which the transfer images are a small set (~20) from the same test space. This is akin to having a few annotated images from the test space and expecting the model to propagate the labels to the rest of the images of the test space (hence the name “label propagation”). In this setting, any remaining gap can be attributed to the suboptimality of pre-training methods, as the transfer set is distribution shift-free.
> > 	- Below are the results of this controlled setup, which paint the same picture as the previous results and provide a reassuring consistency.
> >
> > ```
> >                           MAE    TST-MAE
> >                           31.05  37.98
> > ~Lower Bound ●-----------●●------●-●-----●-----●--------------------------------------------● ~Upper Bound
> >              19.39       30.57     39.81 45.80 51.25                                        91.81
> >              Scratch   Multimodal  TST   4M-21 TST-MM                                       Fully Supervised
> >                        Scratch     Sensors                                                  Large Data
> >
> > Scannet++, ViT, Semantic Segmentation (label propagation)
> > ```
> >
> >
> > References
> >
> > [1] On the Surprising Effectiveness of Attention Transfer for Vision Transformers. Li et al, 2024
> >
> > [2] CC12M, Conceptual 12M: Pushing Web-Scale Image-Text Pre-Training To Recognize Long-Tail Visual Concepts. Changpinyo et al, 2021
> >
> > [3] ScanNet++: A High-Fidelity Dataset of 3D Indoor Scenes. Yeshwanath*, Liu* et al, 2023

---

### Author Response · Authors · 2025-12-03
**AC Summary Comment, part 1**

Dear AC,

Our work studies a paradigm towards a highly performant model for a test (or deployment) space through **specialization** and **multimodality as self-supervision**.

The feedback was strongly positive, and reviewers highlighted the **interesting paradigm and narrative amid the current generalist and scaling-based dominance** (NjHj, ygUd, r7Ks), the **real-world impact** (ygUd, r7Ks), and **investment toward doing analyses** and extracting insights (such as data scaling vs. modality scaling, the specialization-generalization trade-off, etc.) (ygUd, r7Ks).

We provided detailed responses to all reviewers' questions ([NjHj](https://openreview.net/forum?id=4dMlAKBwrA&noteId=Fo7L9USKPv), [ygUd](https://openreview.net/forum?id=4dMlAKBwrA&noteId=Fo7L9USKPv), [r7Ks](https://openreview.net/forum?id=4dMlAKBwrA&noteId=25bNfZ3Nei)), resulting in the following scores before the reversion. We certify that no communication between the authors and reviewers outside the open review system happened, and we are sincerely unaware of the reviewers’ identities:

- `ygUd` adjusted their score to 10, remarking on the work's potential to *"open new avenues and challenges for the community"* ([ygUd](https://openreview.net/forum?id=4dMlAKBwrA&noteId=RK5GQ1UQly)). The reviewer provided an in-depth review and raised points that we totally agreed with. We responded by providing *a consolidated picture to distill and demonstrate the core achievements unambiguously, new further controlled experiments, and a list of steps on how they’ll be reflected in the camera-ready*. This strengthens the paper's presentation, and we’re grateful to the reviewer for it.
- `r7Ks` adjusted their score to 8, acknowledging their concerns being resolved ([r7Ks](https://openreview.net/forum?id=4dMlAKBwrA&noteId=ztT3bO50p3)). We wanted to follow up with them to see if they had any additional concerns we could address to strengthen the paper, since they didn’t raise significant weaknesses, but we didn't get a chance before the intervention in the author reviewer discussion this year.
- `NjHj` (initial score, 4), did not participate in the discussions and post-rebuttal phase, despite our [follow-ups](https://openreview.net/forum?id=4dMlAKBwrA&noteId=B0Rsxaxukp). The raised concerns were minor (e.g., expecting architectural novelty, while the paper's focus is not architecture). We provided a response.

---

> ### Author Response · Authors · 2025-12-03
> **AC Summary Comment, part 2**
>
> We briefly summarize some of the technical discussion with ygUd. Please read our full response [here](https://openreview.net/forum?id=4dMlAKBwrA&noteId=yLEUoxFnuQ).
>
> We present a consolidated picture by plotting all relevant methods on one spectrum below. The lower bound is scratch (i.e., no pre-training and learning using the external transfer set only). The upper bound is approximated by a fully supervised model trained with a large number of annotated segmentation images. The gap between the lower and upper bounds is the playfield for the pre-training methods to fill. Our method is represented as TST (Test-Space Training).
> ```
>                Multimodal   TST-MAE         TST-Sensors
>                Scratch
>                7.82        17.62           28.67
> ~Lower bound ●-●------●----●--------------●●--------●---------------------------------------● ~Upper bound
>              7.49     13.74               27.59     34.49                                   62.03
>              Scratch  MAE                 4M-21     TST-MM                                  Fully Supervised
>                                                                                              Large Data
>
> Scannet++, ViT-B backbone, Semantic Segmentation
> ```
>
>
>
> Key Observations:
> 1. *TST-MM is the most performant model in the test space*. It currently covers half of the way between the bounds. It leverages sensory and off-the-shelf pseudolabelled modalities, with just test space data for pre-training. Note that this model does not qualify for claiming “no external access” (discussed with reviewer ygUD [here](https://openreview.net/forum?id=4dMlAKBwrA&noteId=yLEUoxFnuQ)), as it uses pseudo-labelers trained on external data, akin to distilling their outputs (albeit distilled on test-space data only). This will be reflected in the camera-ready clearly.
> 2. *How far can we go with no external world access and supervision?* To explore this, Section 4.2 presents a reduced case that truly uses no external data, directly or indirectly, as it only uses the sensors on the device to get its modalities. The resulting model, TST-Sensors, covers roughly 1/3 of the way, providing a nontrivial value, and is competitive with internet-based generalist, 4M-21, trained on large-scale internet data (CC12M).
> 3. *Distilling on test-space data vs external data*. The distillation of such pseudo-labelers is more effective on test-space data than on external data, which is intriguing. Comparing TST-MM with 4M-21 shows that, as the two are equivalent in nearly all aspects, except that 4M-21 distills the pseudo-labelers on external data (CC12M) while TST-MM distills the same pseudo-labelers only on test-space data. The same observation was consistently made for various settings in Figure 20 of the Appendix R, and reflections of this underlying trend are also just being observed in recent papers in other areas.
> 4. *Multimodality as supervision is more performant*. Generally, multimodality is useful as the multimodal models consistently outperform their single-modal counterparts (e.g., TST-MM vs TST-MAE or 4M-21 vs MAE).
>
> **Privacy Aspect of Using Test-Space Data.**  We also summarize the discussion on the privacy aspect of our work, which ygUd expressed interest in (a similar discussion can be found in the Ethics Statement, Section 6).
>
> 1. The paper's privacy aspect is interesting. It presents both a risk and an opportunity, and we think the opportunity is greater. Despite the risk of sending out test-space data to be trained on (as noted by ygUd), TST presents a pathway for a specialization setting, which can ultimately enable training a compact user-dedicated model with some in-house compute for sensitive domains, in a *fully local* fashion.
> 2. On the other hand, adopting the currently dominant paradigm of *generalists* trained on large-scale data essentially requires the data across many users to be harvested and pooled to create such training datasets that cover all modes. This poses its own privacy risk — just not for the specific user that deploys the model, but for all the other users whose data had to be harvested to enable the creation of a generalist. It can also incentivize the commodification of people’s data by large companies, which is often undesirable to the users. TST takes a step towards developing an effective *specialization* paradigm that *reduces the need for external data*, thereby lowering the need to pool different users’ data to build large training datasets.

---

### Meta-Review · Area_Chair_ZBFz · 2025-12-31

**Summary:**

* The paper addresses a timely and relevant problem of specialization via test-space training with multimodal self-supervision for deployment scenarios.
* The empirical evaluation is extensive and carefully executed, yielding nontrivial insights (modality scaling vs. data scaling, specialization vs generalization trade-offs, test-space vs external distillation).
* Initial concerns about over-claiming and framing were acknowledged and addressed through clarification and planned revisions.
* A substantive concern remains regarding limited methodological novelty: the framework largely integrates existing components (no new algorithm, loss, or formal learning framework) and the method description is thin.
* Reasons for acceptance: empirical support and relevance and very positive scores from 2 reviewers, although AC views the paper as borderline and shares concerns with the negative reviewer.

**Reviewer Concerns:**

Reviewer NjHj:
* [Partly addressed] The authors clarify the intended contribution, positioning TST as a paradigm and emphasizing multimodality as self-supervision in the test space. This addresses the reviewer’s concern at a high level but does not resolve it.
* [Not addressed] The authors' response downplays the reviewer's concern by reframing it as an expectation of architectural novelty, which the review does not state / ask for. The reviewer instead asks for a stronger methodological contribution. After the rebuttal, the technical method remains thin, and it is still unclear what is technically new.

Reviewer ygUd:
* [Addressed] Over-claiming/framing - authors commit to rewriting abstract/intro/teaser to clarify, reviewer's questions addressed.

Reviewer r7Ks:
* [Addressed] Authors justify that 3-5k eval images are sufficient and show mIoU is stable, fix formatting issues and promise to add the requested explanations.

All reviewers find the work timely and interesting. Two reviewers mention their intention to increase the scores.

What confuses me is that the authors state the reviewer NjHj did not participate "despite follow-ups". Given the discussion freeze on Nov 27 (and messages by the authors were posted on Nov 27), this phrasing is misleading. The reviewer was unable to reply at that point.

**Reviewer Scores:**

* Reviewer NjHj: Would have kept the score --> 4 (from 4) NjHj’s concerns were primarily about methodological novelty and thin technical content. They are not fully addressed by the rebuttal.
* Reviewer ygUd: Would have raised the score --> 8 or 10 (from 6) as ygUd explicitly stated.
* Reviewer r7Ks: Would have raised the score --> 8 (from 6), as r7Ks explicitly stated.

---

### Decision · Program_Chairs · 2026-01-26

Accept (Poster)